

# An intercomparison of approaches for improving predictability
# in operational seasonal streamflow forecasting
Pablo A. Mendoza[1,*], Andrew W. Wood[1], Elizabeth Clark[2], Eric Rothwell[3], Martyn P. Clark[1], Bart
Nijssen[2], Levi D. Brekke[4] and Jeffrey R. Arnold[5]
[1]Hydrometeorological Applications Program, National Center for Atmospheric Research, Boulder, Colorado, USA
[2]Department of Civil and Environmental Engineering, University of Washington, USA
[3]Bureau of Reclamation, Boise, USA
[4]Bureau of Reclamation, Denver, USA
[5]Climate Preparedness and Resilience Programs, U.S. Army Corps of Engineers, Seattle, USA
*Correspondence to*: Pablo A. Mendoza (pmendoza@colorado.edu)
**Abstract.** For much of the last century, forecasting centers around the world have offered seasonal streamflow
predictions to support water management. Recent work suggests that the two major avenues to advance seasonal
predictability are improvements in the estimation of initial hydrologic conditions (IHCs) and the incorporation of
climate information. This study investigates the marginal benefits of a variety of methods using IHC and/or climate
information, focusing on seasonal water supply forecasts (WSFs) in five case study watersheds located in the U.S.
Pacific Northwest region. We specify two benchmark methods that mimic standard operational approaches – statistical
regression against IHCs, and model-based ensemble streamflow prediction (ESP) – and then systematically inter-
compare WSFs across a range of lead times. Additional methods include: (i) statistical techniques using climate
information either from standard indices or from climate reanalysis variables; and (ii) several hybrid/hierarchical
approaches harnessing both land surface and climate predictability. In basins where atmospheric teleconnection
signals are strong, and when watershed predictability is low, climate information alone provides considerable
improvements. For those basins showing weak teleconnections, custom predictors from reanalysis fields were more
effective in forecast skill than standard climate indices. ESP predictions tended to have high correlation skill but
greater bias compared to other methods, and climate predictors failed to substantially improve these deficiencies
within a trace weighting framework.  Lower complexity techniques were competitive with more complex methods,
and the hierarchical expert regression approach introduced here (HESP) provided a robust alternative for skillful and
reliable water supply forecasts at all initialization times. Three key findings from this effort are: (1) objective
approaches supporting methodologically consistent hindcasts open the door to a broad range of beneficial forecasting
strategies; (2) the use of climate predictors can add to the seasonal forecast skill available from IHCs; and (3) sample
size limitations must be handled rigorously to avoid over-trained forecast solutions. Overall, the results suggest that
despite a rich, long heritage of operational use, there remain a number of compelling opportunities to improve the skill
and value of seasonal streamflow predictions.





## 1    Introduction

The operational hydrology community has long grappled with the challenge of producing skillful seasonal
streamflow forecasts to support water supply operations and planning. Proactive water management has become
critical for many regions in the world that are susceptible to water stress associated with the intensification of the
water cycle. Paradoxically, although we have seen important technological advances – including increased computing
power, the broader availability to climate reanalysis, forecasts and reforecasts, and more complex process-based
hydrologic models (Pagano et al., 2016), the skill of operational seasonal runoff predictions in the US, termed water
supply forecasts (WSFs), has shown little or no improvement over time (e.g., Pagano et al., 2004; Harrison and Bales,
2016). Hence, there is both a scientific and practical need to understand the potential of new datasets, modeling
resources and methods to accelerate progress towards more skillful and reliable operational seasonal streamflow
forecasts.
There is general consensus in the research community on the main opportunities to improve seasonal streamflow
prediction skill (e.g., Maurer et al., 2004; Wood and Lettenmaier, 2008; Yossef et al., 2013). These include improving
knowledge of: (i) the amount of water stored in the catchment – hereinafter referred to as initial hydrologic conditions
(IHCs), and (ii) weather and climate outcomes during the forecast period. Our ability to leverage the first predictability
source (i.e., hydrologic predictability) depends on the accuracy of watershed observations and models, including
model input forcings (e.g., precipitation and temperature), process representations, and the effectiveness of hydrologic
data assimilation (DA) methods. Our ability to leverage the second source (climate predictability) depends both on
how well we can characterize and predict the state of the climate and on how effectively we can incorporate this
information into streamflow forecasting methods. This idea has been explored in different frameworks using standard
indices – e.g., Niño3.4, the Pacific Decadal Oscillation (PDO) – and/or custom (i.e., watershed-specific) climate
indices derived from climate reanalyses (e.g., Grantz et al., 2005; Bradley et al., 2015), or using seasonal climate
forecasts to run hydrologic model simulations (e.g., Wood et al., 2005; Yuan et al., 2013).
Despite generally promising findings from this body of work and from a number of agency development efforts
(Weber et al., 2012; Demargne et al., 2014), current operational practice in the US still takes little to no advantage of
large-scale climate information for real-time seasonal streamflow forecasting. Clear examples can be found in the
western United States, a large snowmelt dominated region where official WSFs are produced via two main approaches:
(i) statistical models leveraging in situ watershed moisture measurements such as snow water equivalent (SWE),
accumulated precipitation and streamflow (Garen, 1992; Pagano et al., 2004); and (ii) outputs from the National
Weather Service (NWS) Ensemble Streamflow Prediction method (ESP; Day, 1985; Crochemore et al., 2016), based
on watershed modeling. These approaches rely solely on the predictability from IHCs and do not leverage any type of
large-scale current or future climate state information that might influence the forecasted hydrologic outcomes.
This paper presents an assessment of several seasonal streamflow prediction approaches in harnessing both
watershed and climate related predictability. The methods are applied to seasonal WSFs and span a range of
complexity, from purely statistical to purely dynamical and hybrid statistical/dynamical approaches. In this paper,
'increased complexity' indicates a gradient from purely data-driven techniques (e.g., linear regression) to the use of
dynamical watershed models (Plummer et al., 2009), the outputs of which may be further processed using additional





statistical approaches. Although most of the techniques evaluated here are not new, the intercomparison offers new
insights for researchers and developers in the operational community because: (1) the experiment is broader than prior
efforts and benchmarks alternative methods against current operational ones; and (2) the methods are chosen to be
operationally feasible, avoiding the use of data that cannot be obtained in real-time. In addition, the work uses a
hindcast/verification framework and follows more rigorous standards for cross-validation than were used in some of
the prior studies.
The remainder of this paper is organized as follows. Section 2 describes prior methodological work and context
for statistical, dynamical and hybrid approaches to seasonal streamflow forecasting. The study domain is described in
Section 3. Datasets, experimental design, individual methods, and forecast verification measures are detailed in
Section 4. Results and discussion are presented in Section 5, followed by the main conclusions of this study (Section

81  6).

**2    Background**
Seasonal streamflow forecasting methods can be categorized as dynamical, statistical, or hybrid, and span
different degrees of complexity and information requirements. Dynamical methods use time-stepping simulation
models to represent hydrologic processes. They describe future climate using either historical meteorology or inputs
derived from seasonal climate forecasts (e.g., Beckers et al., 2016). On the other hand, statistical or purely data-driven
methods rely on empirical relationships between seasonal streamflow volumes, and large-scale climate variables
and/or in situ watershed observations. Several statistical approaches can be found in the literature, encompassing
different degrees of complexity (e.g., Garen, 1992; Piechota et al., 1998; Grantz et al., 2005; Tootle et al., 2007; Wang
et al., 2009; Moradkhani and Meier, 2010). Other studies have tested multi-model combination techniques for purely
statistical seasonal forecasts, using objective performance criteria (e.g., Regonda et al., 2006), both performance and
predictor state information (Devineni et al., 2008), and Bayesian model averaging (e.g., Mendoza et al., 2014), among
others.
Hybrid methods strive to combine the strengths from both dynamical and statistical techniques. For instance,
uncertainties in dynamical predictions indicate that dynamical forecasts can benefit from statistical post-processing
(e.g., Wood and Schaake, 2008). One line of research has examined the potential benefits of using simulated watershed
state variables – either from hydrologic or land surface models – as predictors for statistical models (e.g., Rosenberg
et al., 2011; Robertson et al., 2013). Another popular technique consists in incorporating climate information within
ESP frameworks, either deriving input sequences of mean areal precipitation and temperature from current climate or
climate forecast considerations (e.g., Werner et al., 2004; Wood and Lettenmaier, 2006; Luo and Wood, 2008; Gobena
and Gan, 2010; Yuan et al., 2013) – referred to as *pre-ESP* –, or ESP weighting (also referred to as *post-ESP*) based
on climate information (e.g., Smith et al., 1992; Werner et al., 2004; Najafi et al., 2012; Bradley et al., 2015). Werner
et al. (2004) found that the post-ESP method (termed 'trace weighting') was more effective than pre-ESP to improve
forecast skill.
The combination of outputs from different models has also been shown to benefit seasonal hydroclimatic
forecasting (e.g., Hagedorn et al., 2005). Although several studies have demonstrated that statistical multimodel



techniques applied on dynamical models tend to outperform the 'best' single model (e.g., Georgakakos et al., 2004; Duan et al., 2007), fewer insights have been gained on combining statistical or dynamical models in seasonal streamflow forecasting. Recently, Najafi and Moradkhani (2015) tested multimodel combination techniques of different complexities from both statistical and dynamical forecasts, concluding that model combination generally outperforms the best individual forecast model. Many sophisticated seasonal forecasting frameworks can be found in the literature, some of which incorporate DA techniques (e.g., Dechant and Moradkhani, 2011), a topic not discussed here. For this reason, the hydrology community may benefit from a broad assessment of the marginal benefits of choices made in a range of seasonal streamflow forecasting frameworks.

## 3 Study Domain

Our test domain is the U.S. Pacific Northwest (PNW) region (Figure 1), which relies heavily on winter snow accumulation and spring snowmelt to fulfill water needs during spring and summer (e.g., Mote, 2003; Maurer et al., 2004; Wood et al., 2005). We select catchments contributing to five reservoirs: Dworshak (DWRI1), Howard Hanson (HHDW1), Hungry Horse (HHWM8), Libby (LYDM8) and Prineville (PRVO). Two of them – Hungry Horse and Prineville reservoirs – are owned and operated by the U.S. Bureau of Reclamation (USBR), while the rest are operated by the U.S. Army Corps of Engineers (USACE).

The main physical and hydroclimatic characteristics of the case study basins are summarized in Table 1. These basins cover a wide range of runoff efficiencies (from 0.13 at Prineville to 0.78 at Howard Hanson) and dryness indices (from 0.63 at Howard Hanson to 3.83 at Prineville). Relatively high basin-averaged elevations condition a pronounced seasonal temperature pattern, with minimum values below the freezing point between December and February, and maximum temperatures during June-September (not shown). These topographic and hydroclimatic features favor snowpack development in the months October-April, stressing the seasonal behavior of other water storages and fluxes. This is illustrated in Figure 2, including model precipitation (i.e., observed precipitation with a snow correction factor, SCF) and monthly averages of hydrologic variables simulated with the Sacramento Soil Moisture Accounting (SAC-SMA, Burnash et al., 1973) and SNOW-17 (Anderson, 1973) watershed models (see Section 4). Although seasonal precipitation patterns may differ, water starts accumulating in October as snow water equivalent (SWE) and/or soil moisture (SM) in all basins. Increases in SM and runoff in most basins are driven by snowmelt at the beginning of spring with the exception of Howard Hanson, where the bulk of annual streamflow occurs in November-May. Among these basins, Dworshak, Hungry Horse and Libby share similar SWE, soil moisture, and runoff cycles, although precipitation is relatively uniform in the last one throughout the year.

The hydroclimatology of the PNW region is affected by a number of large-scale climate teleconnections. The warm (cold) phase of El Niño Southern Oscillation (ENSO) is typically associated with above (below) average temperatures and below (above) average precipitation during winter (e.g., Redmond and Koch, 1991), and therefore decreased (increased) snowpack (Clark et al., 2001) and spring/summer runoff (e.g., Piechota et al., 1997). The Pacific Decadal Oscillation (PDO; Mantua et al., 1997) – which reflects the dominant mode in decadal variability of SSTs – has also been found a relevant driver for the hydroclimatology of the PNW (e.g., McCabe and Dettinger, 2002). The joint influence of ENSO and PDO on North American climate conditions, snowpack and spring/summer runoff has



been also well recognized and documented (e.g., Hamlet and Lettenmaier, 1999). As a consequence, many authors
have explored the incorporation of large-scale climate information for seasonal streamflow forecasting in the PNW –
using either standard indices (e.g., Hamlet and Lettenmaier, 1999; Maurer et al., 2004), custom indices from reanalysis
fields (e.g., Opitz-Stapleton et al., 2007; Tootle et al., 2007), both (Moradkhani and Meier, 2010), or downscaled
climate forecasts (Wood et al., 2005) – finding improved predictability for lead times longer than 2 months, and
particularly in years of strong anomalies in climate oscillations such as ENSO.

## 4    Approach

### 4.1    Experimental Design

We use several decades of seasonal streamflow hindcasts to assess a suite of methods (Figure 3), focusing on
April-July streamflow (runoff) volume, the most common western US water supply forecast predictand. Probabilistic
(ensemble) WSFs for this period are generated the first day of each month from October to April, in every year of the
hindcast period 1981-2015. For the methods involving statistical prediction, we use a leave-three-out cross validation
at all stages of the forecast process. This procedure is repeated for consecutive 3-year periods (e.g., 1984-1986, 1987-
1989, 1990-1992, etc.), except for the last time window (2014-2015).
The techniques assessed here are categorized as follows. The first group, *IHC-based* methods, includes two
approaches (referred to as *benchmark methods*) – ESP and IHC-based statistical – currently used operationally in the
western U.S. (both harnessing only IHC information), and a very simple ESP post-processor to reduce systematic
biases. A second group, *climate-only* methods, includes statistical techniques harnessing climate information from
two different sources – standard indices (e.g., Niño3.4, PDO, AMO), or variables extracted from the Climate System
Forecast Reanalysis (CFSR; Saha et al., 2010). A third group of *hybrid* or *hierarchical* methods includes subgroups
of techniques that: (i) combine watershed predictors (IHCs) and climate predictors (either indices or CFSR variables)
within a statistical framework, (ii) use climate information to post-process outputs from a dynamical method (i.e.,
ESP), or (iii) combine purely climate-based ensembles with purely watershed-based ensembles.
In operational practice, ESP produces an ensemble of streamflow estimates whereas statistical water supply
forecasting yields a statistical distribution. In this study, we generate ensembles of the final predictand for all methods.
An ensemble size 500 is used – wherein the members are generated through a resampling (in some cases weighted)
of the predictive distributions – except for the ESP and bias-corrected ESP methods, for which 32 members are
generated (i.e., 35 total historical years less the three out of sample test years). In the statistical approaches, seasonal
flows are log-transformed and predictor and predictand data are normalized before training statistical method
parameters or weights. The statistical models were then applied in log-standard-normal space for forecast generation,
and predictands are transformed back to streamflow space.





### 4.2 Forecasting Methods

#### 4.2.1 IHC-based methods

**Ensemble Streamflow Prediction (ESP)**

The traditional ESP method (Day, 1985) relies on deterministic hydrologic model simulations forced with observed meteorological inputs up to the initialization time of the forecast. The approach assumes that meteorological data and model are perfect – i.e., there are no errors in IHCs, and that historical meteorological conditions during the simulation period can be used to represent climate forecast conditions. For hindcast verification purposes, the meteorological input traces associated with forecast years must be excluded.

The hydrology models used in this study were the NWS Snow-17, SAC-SMA and a unit-hydrograph routing model, all implemented in lumped fashion with 2-3 snow elevation zones per watershed. The models were calibrated via an automated multi-objective parameter estimation to reproduce observed daily streamflow. Hydrologic model forcings were drawn from a 1/16 degree real-time implementation of the ensemble forcing generation method described in Newman et al. (2015). Naturalized flow data was obtained from a combination of sources, including the Bonneville Power Administration (BPA, 2011), the USBR Hydromet historical data access system, and the USACE Data Query System.

Figure 4 shows simulated and observed monthly time series of streamflow for the period Oct/1990 – Sep/2000. With the exception of Prineville, where neither meteorology nor flow are well measured, all basins show values of NSE and $r$ higher than 0.76 and 0.87, respectively. Further, the climatological seasonality of streamflow is reproduced well in all basins.

**Statistical forecasting using initial hydrologic conditions (Stat-IHC)**

This method mimics the approach of the U.S. Natural Resources Conservation Service (NRCS), but differs in using model-simulated basin-averaged SWE and SM as surrogates for ground-based observations of SWE, precipitation and streamflow used operationally by the NWS and NRCS (as demonstrated in Rosenberg et al., 2011). A linear regression equation is developed between log-transformed seasonal runoff and IHCs represented by the sum of simulated basin-averaged SWE and SM. The training period equations are used to issue a deterministic runoff volume prediction for each year left out, and ensembles are generated by adding 500 Gaussian random numbers with zero mean and a standard deviation equal to the standard error of the individual prediction. The predictions are then exponentiated.

**Bias Corrected Ensemble Streamflow Prediction (BC-ESP)**

ESP predictions often exhibit a systematic bias due to inadequate model parameters and/or other sources or error (e.g., input forcing selection, model structure). If the ESP approach provides a consistent hindcast, as it does here, post-processing in the form of a simple bias-correction (BC-ESP) can be applied. This is achieved by multiplying the raw ESP forecasts by a mean scaling factor that is obtained by computing the ratio between the mean of observed seasonal runoff volumes (i.e., the predictand) and the mean of ESP forecast median volumes, for each initialization time. Each scaling factor calculation and application is cross-validated.





### 4.2.2 Statistical forecasting harnessing only climate information

**Multiple linear regression (MRL) using standard climate indices (Stat-Ind)**

This method evaluates 12 standard climate indices as candidate predictors (Table 2). For each initialization time (e.g., November 1) and climate index (e.g., Niño3.4), the 3-month time window that maximizes the correlation coefficient between a preceding seasonal (e.g., August-October) predictor average and seasonal streamflow volume over the training period is selected. Once this procedure is repeated for all potential predictors, the best possible time series are obtained for the 12 climate indices, and ensemble forecasts are produced for a given initialization through the following steps:

1. Several combinations of predictors are selected subject to the constraint that no pairs of predictors with an inter-correlation larger than $C_{thresh} = 0.3$ should be included.

2. Stepwise MLR models are fit for all combinations of predictors identified in Step 1, and the set of predictors that minimizes the Bayesian Information Criterion (BIC) score (Akaike, 1974) over the training period is selected.

3. An ensemble forecast is generated (as for Stat-IHC) with the MLR model from Step 2.

We choose MLR over more parameterized regression methods (e.g., local polynomial regression) since these were found to perform poorly in cross-validation, mainly due to the limited samples sizes available in the seasonal hydrologic prediction context.

**Partial Least Squares Regression using reanalysis fields (Stat-CFSR)**

The teleconnections captured in off-the-shelf climate indices are not influential everywhere. Therefore, we also assess the potential of custom climate predictor indices derived from reanalysis data. Following Tootle et al. (2007), we use Partial Least Squares Regression (PLSR; Wold, 1966) to extract information from climate fields. PLSR decomposes a set of independent variables $X$ and dependent variables $Y$ into a small number of principal components that explain as much covariance as possible between the two variable sets (Abdi, 2010). PLSR components are formed from CFSR 700 mb geopotential height (Z700) and sea surface temperatures (SSTs) over the domain 20°S–80°N; 130°E–10°W. For dates beyond 2010, we merged the 1979-2010 CFSR data with monthly analysis fields from the Climate Forecast System version 2 (CFSv2; Saha et al., 2014), aggregating the latter product to 2.0° × 2.0° horizontal resolution. Similar to the Stat-Ind method, we use 3-month averages of these variables. The seasonal forecasts are generated for each initialization by following these steps:

1. Compute principal components from the combined SST and Z700 gridded values for each training sample and the left-out prediction years.

2. Fit a regression model to the resulting PLSR components (predictors), accepting each additional component only when its mean partial correlation with volume runoff is above a threshold. We used a threshold of 0.30 throughout the study after finding that nearby values – e.g., 0.25, 0.35 – did not substantially change the results. The small sample size and low predictability supported at most two components.

3. Compute a mean runoff volume forecast using the regression model obtained in Step 2, and generate an ensemble by adding 500 Gaussian random numbers with zero mean and a standard deviation equal to the





root mean squared error of prediction (RMSEP) obtained from leave-three-out cross validation within the

training period.

The main implication of developing PLSR components and the subsequent estimation of regression coefficients

in cross validation – as conducted here – is that climate information from the target prediction period is not used at
all, as is the case in real-time systems. This is a key methodological difference versus past studies that used all
historical available information to define custom reanalysis predictor fields (e.g., Grantz et al., 2005; Regonda et al.,
2006; Bracken et al., 2010; Mendoza et al., 2014), yielding a moderate yet erroneous boost in predictability.

### 4.2.3    Hybrid/hierarchical methods combining watershed and climate information

**Stepwise MLRs using IHCs and climate predictors**

We applied two statistical methods that combine climate and dynamical watershed model predictors: Stat-Ind-

IHC (which uses climate indices and IHCs), and Stat-CFSR-IHC (which uses CFSR-based PLSR components and
IHCs). These approaches are implemented in identical fashion to Stat-Ind, except that IHCs are added to the potential
suite of climate predictors.

**Hierarchical Ensemble Streamflow Prediction (HESP)**

The underlying idea of HESP is that the two main sources of predictability – watershed IHCs and climate – may

best be addressed sequentially to ensure that only climate uncertainty is related to climate predictors. This may not the
case if a climate variable enters first into a regression model that attempts to explain streamflow variance from both
IHCs and climate, possibly leading to a sub-optimal predictor selection. HESP is thus a hierarchical regression
approach in which streamflow is first related to IHCs by fitting $Q = f$(IHC predictors) $+ \varepsilon_{climate}$, given sufficient IHC
predictor strength. The residual uncertainty is then related to climate predictors (again if possible) by fitting $\varepsilon_{climate}$
$= g$(climate predictors) $+ \varepsilon_{residual}$, such that the final forecast equation takes the form:
$Q = f(\text{IHC predictors}) + g(\text{climate predictors}) + \varepsilon_{residual}$                  (1)

Here the predictor pool used to explain $\varepsilon_{climate}$ may include standard climate indices or reanalysis PLSR

components, depending on the performance obtained during the training period. Absent IHC predictability, HESP is
equivalent to Stat-Ind or Stat-CFSR; whereas without climate predictability, it defaults to Stat-IHC. Lacking both IHC
and climate predictability, HESP defaults to climatology – i.e., an ensemble forecast is issued by resampling from
historical observations over the training period.

**ESP Trace Weighting Scheme (TWS)**

A well-known strategy for incorporating climate information into ESP forecasts is called 'trace weighting'

(Smith et al., 1992; Werner et al., 2004), where forecasted flow probabilities are corrected by weighting each ensemble
member according to the similarity between a climate-related feature of the current year (e.g., PDO) and the
meteorological year used to generate that member. Here, for a given basin and forecast period, either climate indices
or CFSR-based components are selected based on their training period performance (i.e., RMSE) and used to weight
each trace obtained from BC-ESP (see Section 7.1 for further details).




**Equally weighted ensembles (EWE) and RMSE-weighted ensembles (RWE)**

EWE combines the best-performing climate-only hindcast (i.e., Stat-Ind or Stat-CFSR, based on RMSE over the training period) with the best watershed-only hindcast (either Stat-IHC or BC-ESP), resampling ensemble members equally from each source to form a new 500-member ensemble forecast. A variation of this combination approach (RWE) instead performs a weighted resampling from the two forecast sources according to their skill during the training period: i.e., the weights equal 1/RMSE, where RMSE the root mean squared error of the ensemble median.

**Bayesian Model Averaging (BMA) and Quantile model averaging (QMA)**

These methods combine the best-performing climate-only hindcast with the best performing watershed-only hindcast. While BMA (Raftery et al., 2005) attempts to provide a weighted average of forecast probability densities, QMA (Schepen and Wang, 2015) applies a weighted average to forecast values (quantiles) for a given cumulative probability. A notable difference between the two approaches is that QMA produces smoother and consistently unimodal distributions compared to potentially bimodal BMA outputs (Schepen and Wang, 2015). More details on these techniques are provided in section 7.2.

### 4.3 Forecast evaluation

Forecast method performance was evaluated using the metrics listed in Table 3. These include some standard metrics used in hydrology, such as correlation coefficient ($r$), root mean squared error ($RMSE$), and percent bias, and also probabilistic measures to assess skill and reliability. Skill is obtained using the continuous ranked probability score (CRPS; Hersbach, 2000), which measures the temporal average error between forecast CDF with that from the observation. Forecast reliability – i.e., adequacy of the forecast ensemble spread to represent the uncertainty in observations – is evaluated using an index from the predictive quantile-quantile (QQ) plot (Renard et al., 2010). QQ plots compare the empirical CDF of forecast $p$-values (i.e. $P_i(o_i)$, where $P_i$ and $o_i$ are the forecast CDF and observation at year $i$) with that from a uniform distribution $U[0,1]$ (Laio and Tamea, 2007).

Confidence intervals for the verification statistics are created using bootstrapping with replacement. In each resampling step, $N$ pairs of ensemble forecasts and observations were resampled from the original joint distribution ($N$ is the total number of events for which probabilistic forecasts are available). This process is repeated 1000 times, and all statistics are then computed for each realization and ranked in order to obtain 95 % confidence limits.

### 5 Results and discussion

### 5.1 Deterministic evaluation

We first compare methods using the WSF median, a critical predictand for many water decisions (e.g., Lake Powell releases on the Colorado River in the western US). Figure 5 displays correlation coefficients ($r$) between forecast median and observed April-July runoff volumes for the five case study basins. As expected, near-zero or negative $r$ values were obtained for October 1 and November 1 WSFs with the IHC-based methods. Negative correlation scores arise in very low-skill situations as an artifact of cross-validation (e.g., leaving a high predictand out of a training sample biases the resulting prediction in the opposite direction). The seasonality of SM and SWE in





the basins of interest (Figure 2) does not yield watershed moisture accumulations with predictive power until
December or January. In contrast, $r$ values as high as 0.48 for Dworshak and 0.49 for Hungry Horse could be attained
on October 1 using only information from climate indices (Stat-Ind). Generally, but not everywhere, methods
harnessing predictability from the climate (with the exception of TWS) enhance skill in comparison to IHC-based
methods at initializations early in the water year. TWS is unable to shift the parent ESP distribution sufficiently to
impart much climate skill at this time of year.
After January, the hydrologic model begins to capture a useful moisture variability signal from the watershed,
thus IHCs start to become a dominant source of predictability in all basins. Indeed, watershed information is
particularly relevant at Libby and Prineville (Figure 5d and 5e), where correlations within the range 0.39-0.47 are
achieved as early as December 1 with the three IHC-based techniques. In these basins, standard climate indices do not
provide useful long-lead predictability, although CFSR-based predictors do support a consistent improvement. For
example, the correlation from Stat-Ind for Libby (Prineville) on December 1 is -0.23 (0.02), while the $r$ value from
Stat-CFSR is 0.19 (0.30). These differences between Stat-Ind and Stat-CFSR remain at these basins for subsequent
monthly initializations.
Figure 5 reveals several notable outcomes that are evident in many of the results plots. First, a linear regression
against IHCs can provide similar $r$ values than the more computationally expensive ESP method, especially at late
initializations (i.e. March 1 or April 1). Likewise, straightforward ensemble combination techniques (e.g., EWE or
RWE) may outperform more complex methods such as BMA (e.g., February 1 – April 1) at all basins. From a
correlation skill perspective, on the other hand, ESP generally outperforms the rest of the methods in late winter and
spring. For example, ESP provides the highest $r$ values for Dworshak (0.82) and Howard Hanson (0.67) on April 1.
Notably, EWE was found to be the best method on April 1 for Hungry Horse ($r = 0.88$) and Prineville ($r = 0.79$) based
on correlation. This indicates that, although simple post-processing can provide substantial forecast improvement, the
small sample size available for training during the cross-validation process results in noisy parameter estimates that
can undermine the potential correlation skill achievable with techniques that are more complex.
Root mean squared errors (RMSE) for ensemble forecast medians (Figure 6) show that despite some
discrepancies between techniques, inter-method differences are not as large as for correlation. In most basins, errors
can be reduced at earlier initializations (i.e., Oct 1 and Nov 1) by introducing climate information. For instance, on
October 1, Stat-Ind and Stat. Ind+IHC generate respective reductions in RMSE of 10% and 13% at Dworshak, 23%
and 16% at Howard Hanson, and 14% and 12% at Hungry Horse, relative to the best IHC-based method in each basin.
These benefits are seen in most initializations and catchments except at Libby, where the best results were mostly
achieved using ESP (Oct 1) and Stat-IHC (Dec 1, and Feb 1 – Apr 1). In agreement with Beckers et al. (2016), this
study was unable to find encouraging climate teleconnections at Libby, despite its relative proximity to Hungry Horse.
Figure 6 underscores that from a median error perspective, intuitive ensemble combinations approaches (i.e.,
EWE and RWE, shown in dark green) can be effective for reducing forecast errors once the watershed begins to
provide useful predictability (i.e. after January 1). For instance, EWE was the best performing method in Hungry
Horse and Prineville for forecasts initialized on March 1 and Apr 1. Further, Figure 6 illustrates that the best (or worst)
techniques when looking at RMSE vary with each basin, although it is clear that TWS and only-climate methods





perform poorly at early and late initializations, respectively. The joint inspection of Figures 5 and 6 shows that inter-
method agreement in correlation does not necessarily translate into similar forecast median errors. For example, while
ESP and HESP provide close $r$ values at Dworshak (0.74 and 0.73) on March 1, larger discrepancies are obtained in
RMSE, with values of 0.58 million-acre-feet (MAF) and 0.79 MAF for ESP and HESP, respectively.
Another interesting result is that no substantial reductions in RMSE were achieved at Howard Hanson between
October 1 and April 1, in contrast to the gradual growth of hydrologic predictability to support forecast skill in other
basins. Indeed, the best performing techniques for October 1 (Stat-Ind) and April 1 (BC-ESP) forecasts provide similar
RMSE values (~0.064 and 0.065 MAF, respectively). This outcome can be attributed to the relatively more rainfall-
dominated hydrograph of Howard Hanson in comparison to the rest of the catchments (Table 1; Figure 2), and
sustained runoff variability generated by seasonally high SM and fall-winter precipitation.
Figure 7 (forecast median bias) shows that raw ESP outputs have the largest biases through most initializations
at Howard Hanson, Libby and Prineville. In particular, absolute biases at Prineville – which is the worst simulated
basin in the group – increase to 53% on October 1 before decreasing to 20% on April 1. Further, relatively large biases
(in comparison to the rest of techniques) were obtained at late initializations in Dworshak and Hungry Horse.
Excepting Prineville, inter-method differences were not substantial, and none of the methods exceeded a 16% bias at
any initialization. The simple bias correction applied in this study was able to reduce absolute biases to less than +/-
3% at Prineville, and less than +/-1% at the rest of the basins. Hence, from a bias reduction perspective, BC-ESP was
the best technique for most basins/initializations, with the exceptions of Dworshak on Feb 1 and Prineville on Mar 1
and Apr 1, for which Stat. CFSR+IHC and TWS provided the best results.
**5.2    Probabilistic verification**
Figure 8 displays continuous ranked probability skill scores computed with mean climatology as a reference
(CRPSS$_{clim}$). Consistent with the correlation analysis results (Figure 5), better skill values are obtained for long lead
times (i.e. Oct 1 and Nov 1) if climate predictors are incorporated in the forecasting framework. For example, Stat.
(Ind+IHC) augments skill by 56% in HHDM1 and 7% in Hungry Horse with respect to Stat-IHC (i.e., the best
benchmark in terms of CRPSS$_{clim}$) when forecasts are initialized on November 1. The skill of IHC-based methods
generally increases from October 1 to April 1. Nevertheless, at late initializations it is still possible to outperform these
techniques in some basins (e.g., Stat (CFSR+IHC) and EWE in Hungry Horse provide skill increases of 7% and 5%
in April 1 forecasts over the best IHC-based technique). For late season initializations – when IHC predictability is
strong –, it is expected that climate-only forecasts underperform other methods.
The results from Figure 8 corroborate several findings alluded to in Section 5.1. Climate predictors applied to
low-skilled (BC-)ESP forecasts in a TWS framework are less effective than when applied in a separate statistical
method. Additionally, less complex multi-model schemes can perform better than more complex approaches (e.g.,
BMA), supporting previous findings by Najafi and Moradkhani (2015). Among the three hybrid regression methods
(Figure 3), Stat-CFSR-IHC was in most cases the worst performer. This result may be determined by the relative
strength of standard (in particular ENSO) indices for the PNW region. Namely, there is less of an opportunity for
custom predictor components to fill a climate influence gap, and the parameter estimation cost of the CFSR-PLSR





relative to an off-the-shelf index may be more exposed. It should also be noted that skill results – especially those
making use of ESP output – are subject to large uncertainties due to limited sample size (i.e., only 35 years for forecast
generation and verification).
Overall, the results presented in Figures 5 and 8 suggest a division of the study basins into two groups showing
different relative predictabilities – i.e., driven by watershed conditions versus climate – from October to January. The
first group is formed by Dworshak, Howard Hanson and Hungry Horse, where the state of the climate is the dominant
source of predictability from Oct 1 to Dec 1, and IHCs start providing useful information on Jan 1. The second group
is formed by Libby and Prineville, where little or no skill can be found from any source until Dec 1, when some
predictability can be harnessed from IHCs. This is illustrated in Figure 9, where time series with cross-validated
seasonal streamflow forecasts – initialized on December 1, period 1981-2015 – are shown for two IHC-based methods
(BC-ESP and Stat-IHC), and two climate-based statistical methods (i.e. Stat-Ind and Stat-CFSR). At such
initialization, there is not enough information in the watershed (IHCs) to predict interannual variations in April-July
streamflow at Dworshak (Figure 9a) or Howard Hanson (Figure 9b); nevertheless, climate predictors are more
successful, a result that is also reflected in positive correlation results (Figure 5) and skill scores (e.g., $CRPSS_{clim}$
increases from 0.23 with Stat-IHC to 0.39 with Stat-Ind at Howard Hanson). For the particular case of Hungry Horse
(Figure 9c), some predictability is provided by watershed information alone (i.e., BC-ESP), although with smaller
correlation and skill than Stat-Ind or Stat-CFSR. Finally, the ensemble forecast time series displayed for Libby (Figure
9d) and Prineville (Figure 9e) portray the relative predictive power of IHCs in these basins compared to climate
predictors alone. Indeed, at the December 1 initialization in these basins, watershed information alone supports $r$
values of 0.43 (Libby) and 0.39 (Prineville) from BC-ESP, and $r$ values of 0.47 from Stat-IHC.
Forecast reliability can be critical to support risk-based decision making, in which actions may be tied to the
forecast distribution rather than the median. The reliability index α (Figure 10), which measures the closeness between
the empirical CDF of forecast $p$-values with a theoretical CDF of $U[0,1]$ (Table 3) shows that – although (BC-)ESP
forecast correlation (Figure 5) and skill (Figure 8) generally increase during the year, forecast reliability from the ESP
methods degrades (i.e. toward lower α) as the initializations approach Apr 1. Because TWS is constrained by ESP
spread, it cannot provide substantial enhancements to poor late-season reliability indices obtained with (BC-)ESP.
In general, forecasts involving statistical calibration (which helps to improve spread and bias) are most reliable.
Indeed, regression-based forecasting methods (e.g., Stat-IHC, Stat-Ind, Stat. Ind+IHC) stand out in all basins,
suggesting that the ensemble generation approach used in this paper (based on the standard error of the cross-validated
hindcasts) is capable of providing statistically consistent ensembles. Multi-model techniques appear to inherit this
characteristic, with only small discrepancies apparent between them (green lines in Figure 10). Similar inter-method
differences across multiple initializations were found when looking at the ε reliability index (not shown) defined by
Renard et al. (2010).
Although HESP was only found to be the 'most reliable' method in a limited number of cases (e.g., α = 0.95 at
Dworshak on Oct 1; α = 0.96 at Libby on Apr 1), relatively high α values were consistently attained in all basins and
forecast lead times. This suggests – in conjunction with the results shown in Figures 5-8 – that HESP has strong
potential for operational streamflow forecasting at all initialization dates, since it is capable of flexibly harnessing



seasonally varying sources of predictability. Figure 11 illustrates this idea through time series of cross-validated ensemble forecasts obtained with HESP for three initialization times (Oct 1, Jan 1, and Apr 1). Forecasts issued on Oct 1 provide positive skill with respect to climatology in Dworshak, Howard Hanson and Hungry Horse, and although CRPSS relative to ESP does not necessarily improve, the associated correlation coefficients (0.42, 0.37 and 0.47, respectively) are a clear enhancement over negative $r$ values obtained from IHC-based methods. The lower probabilistic skill and near-zero correlation in Libby and Prineville reflect the lack of predictability from either the watershed or climate conditions at such a long lead time. Higher values of $CRPSS_{clim}$ for ensemble forecasts initialized on Jan 1 and Apr 1 reflect the increasing power of IHCs, while smaller (and sometimes negative) $CRPSS_{esp}$ values in some basins reflect the increasing difficulty to outperform ESP as IHCs provide more forecast signal. Overall, HESP provides positive skill with respect to mean climatology in all cases, relatively high $r$ values, and statistically consistent forecast ensembles.

### 5.3 Wet/dry year forecasts

Summary statistics provide an overview of forecast performance, but additional insights can be gained from exploring extreme years in the record – in which forecasts can have disproportionate value to help water managers negotiate atypical challenges – and from visualizing the behavior of the forecasting methods as individual seasons progress. We therefore performed a retrospective comparison of all techniques for two regionally wet (1997 and 2011) and dry (1987 and 2001) water years at Hungry Horse (Figure 12), one of the most teleconnected basins in our study domain. Figure 12 illustrates how SWE and SM, the primary sources of predictability for IHC-based methods, progressively gain influence on ensemble forecasts (e.g., HESP and TWS outputs) as the beginning of the snowmelt season approaches (i.e. April 1). These single-year forecast evolution plots highlight the contrast for late season (i.e. Feb 1 onwards) between overconfident predictions exhibiting poor reliability (e.g., ESP, BC-ESP, TWS), and under-confident forecasts (e.g. EWE and RWE).

Figure 12a,b show that climate information is required to reduce forecast errors in wet years at very long lead times (i.e., Oct 1 and Nov 1), either alone or combined with watershed information through hybrid approaches. For example, the technique that provided the smallest forecast median error on Oct. 1 1997 was TWS. For shorter lead times (i.e., forecasts initialized on March 1 or Apr 1) and WY 1997, the incorporation of IHCs helps to provide a better match with observations compared to methods that only use climate information. Interestingly, reanalysis fields at Hungry Horse provide considerable predictive power for WY 2011 (Figure 12b) at short lead times (e.g., Stat-CFSR provides a forecast median error of 2.7 % on March 1).

In the two dry years, Figure 12c illustrates that climate predictors alone had considerable predictive power at long lead times (i.e., Oct 1 and Nov 1) in WY 1987. However, this was not the case for WY 2001 (Figure 12d), when the method providing smallest forecast median volume errors at all initialization times (i.e., either BC-ESP or TWS) always required knowledge on watershed moisture conditions. This was also the case for other pilot study basins (not shown).

The above results suggest that despite the value of large-scale climate information for this study domain, enhanced hydrologic predictability is critical for accurate streamflow volumes in snowmelt-dominated regions under




extreme climatic conditions, especially during dry years. Past and ongoing efforts aimed to improve basin-scale
meteorological forcing datasets, pursue realistic process representations in hydrologic models, advance parameter
calibration, and improve DA techniques for better IHC estimates have built a robust platform to accelerate progress
in this area. However, a long-term retrospective implementation (that is consistent with the real-time deployment) of
these various modeling decisions and sources of information is critical to understand their performance, and
benchmark methodological choices.
**6     Conclusions**
Generating accurate water supply forecasts is an ongoing challenge for improving water resources operations
and planning. Despite substantial work on seasonal streamflow forecasting methods applied worldwide, the marginal
value of increased complexity and combining different sources of information via different strategies has not been
systematically assessed. In this paper, we compare a range of techniques that leverage predictability from watershed
hydrologic conditions and/or large-scale climate information. The forecast intercomparison showed that hybrid
techniques that leverage hindcasts to combine both sources of predictability could lead to improved skill compared to
current operational approaches. Additional key findings that may be relevant beyond the study domain – due to the
inclusion of both teleconnected and non-teleconnected basins – are as follows:
• In basins showing strong teleconnections between large-scale climate and local meteorology, the use of large-
scale climate information can be an effective strategy to improve seasonal streamflow predictability,
potentially providing skillful forecasts at times when watershed predictability is limited.

• Standard climate indices provide useful information, and custom climate predictors from reanalyses were
also an effective complementary strategy for extracting the signal from climate fields (e.g., SST and
geopotential height).

• The relative importance of watershed IHC versus climate information to predict streamflow was found to
vary even within a small region, depending on sub-domain catchment hydroclimatological characteristics.

• The ESP trace weighting method only provided promising results at forecast lead times where ESP raw
forecasts contained moderate skill, indicating that climate information cannot adequately shift the prior ESP
forecast if it lacks forecast resolution or contains significant bias.

• Increasing methodological complexity does not necessarily translate into better ensemble forecast quality
(e.g., Stat-IHC versus BC-ESP; EWE versus BMA), in part because the small sample sizes associated with
seasonal hindcasts preclude reliable parameter estimation for more elaborate methods. There can be a trade-
off between improving one forecast characteristic (e.g., bias) and degrading another (e.g., correlation skill).

• Cross-validation is an essential part of seasonal forecast development and implementation, particularly where
multiple predictions may be combined based on their purported relative strengths and predictive uncertainty
must be accurately estimated. In the small-sample context of seasonal streamflow prediction, cross-validation
reveals significant limitations in the supportable complexity of statistical forecasting elements.





The often equivocal comparison of methods through multiple verification metrics (e.g., correlation, reliability) for individual wet and dry years, and for different basins, starkly illustrated the challenge of selecting a single method that will provide optimal results for all forecast initialization dates. There is a significant tension between optimizing forecast qualities through a mixture of methods and data sources that vary seasonally and across basins, and an oft-stated preference from forecasters and users for a consistent forecasting methodology. With this in mind, we developed HESP as a flexible data-driven framework to harness skill across varying predictability regimes, although it admittedly departs from the constraint of predictor uniformity.

A notable omission from this intercomparison study is the derivation of climate predictors from global climate model forecasts, a strategy that has also been pursued in this context (e.g., see Crochemore et al. 2016). The experiment summarized here did assess the skill of CFSv2 9-month climate forecasts at an earlier stage, but such evaluation has been excluded from this paper because the results did not show significantly higher skill from the CFSv2 forecasts than the CFSR-based empirical predictions, as is consistent with prior skill assessments (e.g., Yuan et al., 2011). Nonetheless, the topic of augmenting hydrologic predictability from dynamical climate forecasts remains an appealing area for future study and comparison, as does the potential for including IHC data assimilation to enhance watershed model-based predictability (e.g., Dechant and Moradkhani, 2011; Huang et al., 2016). Future work can also explore alternative methodological choices such as multiple hydrological models, different climate datasets or smaller details such as alternative variable transformations in statistical approaches (e.g., Wang et al., 2012).

Finally, this work is part of a larger project that explores the potential of an automated (i.e. 'over-the-loop') forecasting workflow as a viable strategy for operational streamflow prediction that can open the door to potential scientific and technical advances in streamflow forecasting (Pagano et al., 2016). In this context, a critical lesson is that the entire study, in particular the assessment of approach alternatives, depends on the automation of the forecast workflow to enable the generation of hindcasts that are consistent with real-time forecasts. Demonstrating that such over-the-loop methods – all of which were implemented in real-time by the authors during the study period (2015-2017) – can yield credible predictions should be regarded as a strong argument for exploring this objective paradigm in real-world operational agency settings.

## 7   Appendix

### 7.1   ESP trace weighting

The trace weighting scheme used here involves the following steps (Werner et al., 2004):

1. Compute a vector **D** of distances between the vector with climate predictors for the target water year ( $x_t$ ), and the vectors with predictors for the training period ( $x_i$ ):

$$D = (d_1, d_2, \ldots, d_n) \tag{A1}$$

$$d_i = \|x_t - x_i\| \tag{A2}$$

2. Sort the vector **D** from lowest to highest:

$$\tilde{D} = (d_{(1)}, d_{(2)}, \ldots, d_{(n)}), \; d_{(1)} \leq d_{(2)} \leq \cdots \leq d_{(n)} \tag{A3}$$

3. Compute weights using the following equation:



$$w_i = \left[1 - \frac{d_{(i)}}{d_{(k)}}\right]^{\lambda}, \ d_{(i)} \leq d_{(t)}$$ (A4)
$$w_i = 0, \qquad d_{(i)} > d_{(t)}$$ (A5)
$$k = NINT\left(\frac{n}{\alpha}\right)$$ (A6)
where $\lambda$ is a distance-sensitive weighting parameter, $\alpha$ is a parameter that influences the $k$ nearest neighbors
used, and NINT refers to the nearest integer operator. In this paper, we set $\lambda = 2$ and $\alpha = 1$ after conducting
several experiments (not shown).
4.  Normalize weights and construct a cumulative distribution function (CDF) based on these values and the

ESP hindcast.

5.  Resample from the CDF obtained in step 4 using 500 uniform random numbers.
**7.2    BMA and QMA**
The principle of BMA (Raftery et al., 2005) is that given an ensemble forecast with $M$ members, each ensemble
member $f_i$ ($i = 1,2,...,M$) is associated with a conditional PDF $h_i(y|f_i)$, which can be interpreted as the PDF of the
variable $y$ given $f_i$. Thus, the BMA predictive model is:
$$p(y|f_1, ..., f_M) = \sum_{i=1}^{M} w_i h_i(y|f_i)$$ (A7)
where the BMA weight $w_i$ is the posterior probability of forecast $i$ and is obtained based on its relative
performance during the training period. Therefore, the weights $w_i$'s are nonnegative and add up to 1, i.e. $\sum_{i=1}^{M} w_i = 1$
(Raftery et al., 2005).
In this paper, the weights for the two models (best climate-based and best watershed-based) are estimated by
maximum likelihood, assuming that the conditional PDFs of log(Q) are approximated by a normal distribution. The
likelihood is maximized using the expectation-maximization (EM) algorithm (Dempster et al., 1977) which is
implemented     in     the     R     package     ensembleBMA     (https://cran.r-
project.org/web/packages/ensembleBMA/ensembleBMA.pdf)   at   the   public   domain   statistical   software   R
(http://www.rproject.org/). Prior information (i.e., initial weights) is provided by weights computed as 1/RMSE.
Finally, the BMA forecast ensemble is obtained by sampling a fraction of members from each model equal to the
weight $w_i$.
The quantile model averaging (QMA) forecast values are obtained from the weighted average of forecast
quantiles from all models. Schepen and Wang (2015) recently found that nearly identical skill results can be obtained
with BMA and QMA, and that very similar performance can be achieved either by calibrating QMA weights or by
using BMA weights within a QMA framework. Therefore, we obtain the QMA forecast using the same weights
obtained from the BMA calibration, by sorting the ensemble members from the best climate and best watershed
forecast approaches, and computing the weighted average of equally ranked ensemble members from the two sources.





**8    Acknowledgments**
This work was supported through a contract with the U.S. Army Corps of Engineers, and through a Cooperative
Agreement with the U.S. Bureau of Reclamation.

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





**10   List of Figures**







**Table 1: List of basin characteristics. Hydrologic variables correspond to the period October 1980 to September 2015. P, R, PE, RR, and DI denote basin-averaged mean annual values of precipitation, runoff, potential evapotranspiration, runoff ratio, and dryness index, respectively.**

|  | Dworshak | Howard Hanson | Hungry Horse | Libby | Prineville |
|---|---|---|---|---|---|
| Symbol | DWRI1 | HHDW1 | HHWM8 | LYDM8 | PRVO |
| Area (km²) | 6300 | 570 | 4200 | 23270 | 6825 |
| Basin average elevation (m.a.s.l.) | 1290 | 905 | 1773 | 1648 | 1301 |
| Mean annual precipitation, P (mm/yr) | 1182 | 1890 | 1043 | 813 | 349 |
| Mean annual runoff, R (mm/yr) | 761 | 1483 | 676 | 408 | 47 |
| Mean annual PE* (mm/yr) | 1362 | 1191 | 1272 | 990 | 1338 |
| Mean annual RE (R/P) | 0.64 | 0.78 | 0.65 | 0.50 | 0.13 |
| Mean annual DI (PE/P) | 1.15 | 0.63 | 1.22 | 1.22 | 3.83 |

*Potential evapotranspiration using the Priestley-Taylor method

**Table 2: List of climate indices included as potential predictors**

| Index | Pattern |
|---|---|
| Niño 3.4 | East Central Tropical Pacific sea surface temperature (SST) |
| Niño 1+2 | Extreme Eastern Tropical Pacific SST |
| Niño 3 | Eastern Tropical Pacific SST |
| Niño 4 | Central Tropical Pacific SST |
| AMO | Atlantic Multidecadal Oscillation |
| NAO | North Atlantic Oscillation |
| PDO | Pacific Decadal Oscillation |
| PNA | Pacific North American Index |
| SOI | Southern Oscillation Index |
| MEI | Multivariate ENSO index |
| WP | Western Pacific Index |
| TNA | Tropical Northern Atlantic Index |





**Table 3: Performance metrics used to assess and compare seasonal streamflow forecasting methods.**

| Notation | Name | Equation | Description |
|---|---|---|---|
| $r$ | Correlation coefficient | $r = \dfrac{\sum_{i=1}^{N}(q_{m,i}-\overline{q_m})(o_i-\overline{o})}{\sqrt{\sum_{i=1}^{N}(q_{m,i}-\overline{q_m})^2}\sqrt{\sum_{i=1}^{N}(o_i-\overline{o})^2}}$ | Deterministic metric that varies [-1,1] with a perfect score of 1. It measures the linear association between forecasts and observations independent of the mean and variance of the marginal distributions. |
| %Bias | Percent bias | $\%Bias = \dfrac{\sum_{i=1}^{N}(q_{m,i}-o_i)}{\sum_{i=1}^{N}o_i}\times 100$ | Deterministic metric that varies (-∞, ∞), with perfect score of 0. It measures the difference between the mean of the forecasts and the mean of observations. |
| RMSE | Root mean squared error | $RMSE = \sqrt{\dfrac{1}{N}\sum_{i=1}^{N}(q_{m,i}-o_i)^2}$ | Deterministic metric that varies [0,∞), with perfect score of 0. |
| CRPSS | Continuous ranked probability skill score | $CRPSS = 1 - \dfrac{CRPS_{fcst}}{CRPS_{ref}}$ $CRPS = \dfrac{1}{N}\sum_{i=1}^{N}\int_{-\infty}^{\infty}\left[F(q)-F_o(q)\right]^2 dq$ $F_o(q) = \begin{cases} 0, & q < o \\ 1, & q \geq o \end{cases}$ | Probabilistic metric that varies (-∞,1], with perfect score of 1. It measures the skill of CRPS relative to a reference forecast (Hersbach, 2000). CRPS quantifies the difference between the cumulative distribution (CDF) function of a forecast ($F$), and the corresponding CDF of the observations ($F_o$). |
| α | $\alpha$ reliability index | $\alpha = 1 - 2\left[\dfrac{1}{N}\sum_{i=1}^{N}\left|P_i(o_i)-U(o_i)\right|\right]$ | Probabilistic metric that varies [0,1]. It quantifies the closeness between the empirical CDF of sample p-values with the CDF of a uniform distribution. A value of 0 is the worst, and 1 reflects perfect reliability (Renard et al., 2010). |

$q_{m,i}$ : Forecast ensemble median for year $i$.
$\overline{q_m}$ : Temporal average over forecast ensemble medians.
$o_i$ : Observation for year $i$.
$\overline{o}$ : Temporal average of observations.
$P_i(o_i)$ : Non-exceedance probability of $o_i$ using ensemble forecasts at year $i$.
$U_i(o_i)$ : Non-exceedance probability of $o_i$ using the uniform distribution $U[0,1]$.






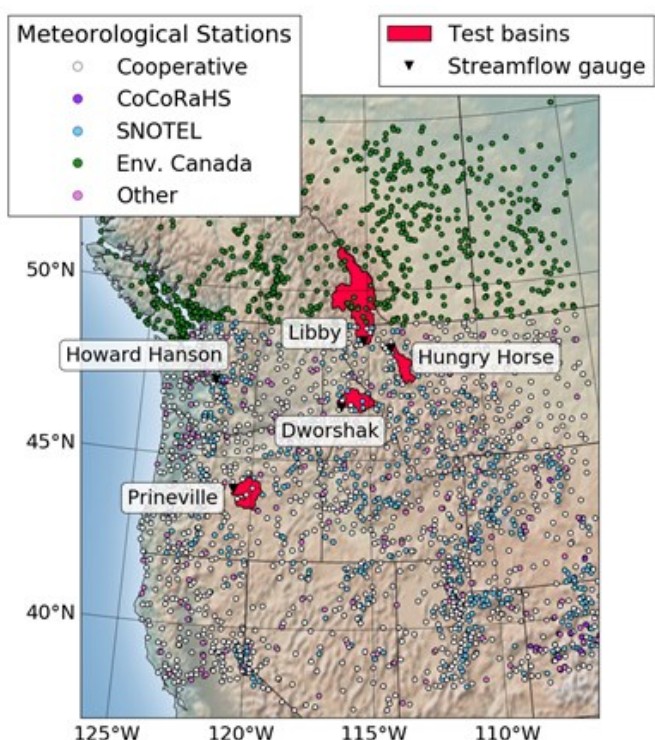

**Figure 1: Location map with the pilot basins included in this study.**



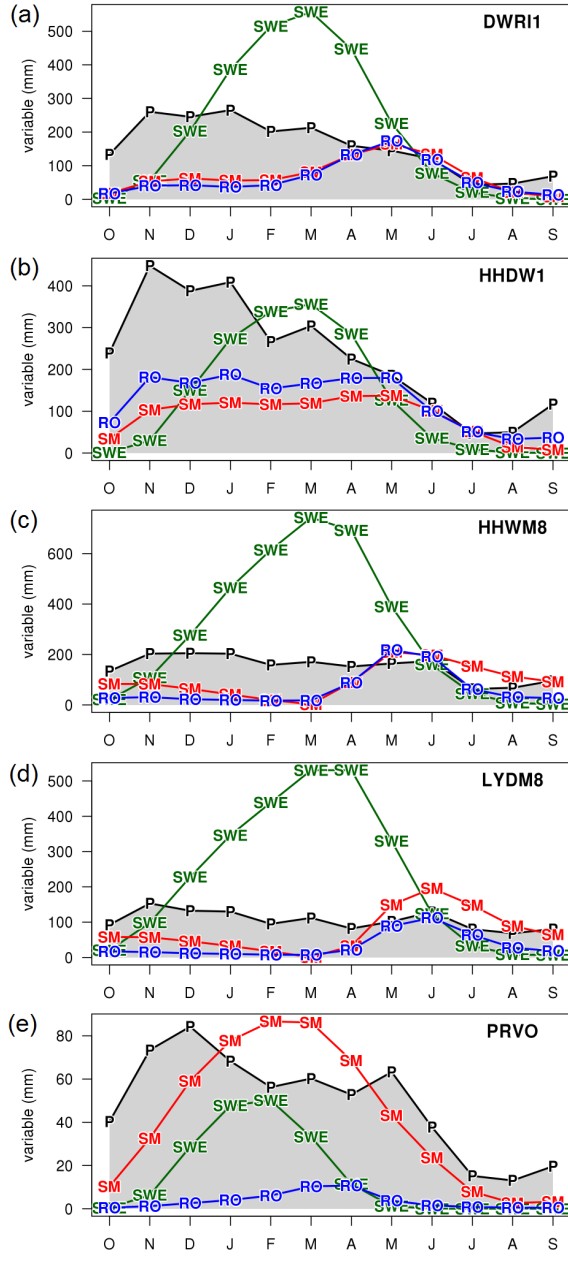


**Figure 2: Corrected precipitation P (i.e. observed precipitation multiplied by a snow correction factor SCF) and simulated water balance variables—active SM, SWE, and runoff (RO)—for the five study basins: (a) Dworshak Reservoir inflow (DWRI1), (b) Howard Hanson reservoir inflow (HHDW1), (c) Hungry Horse reservoir inflow (HHWM8), (d) Libby dam inflow (LYDM8), and (e) Prineville reservoir inflows (PRVO). For model SM, we subtract the lowest mean monthly value of the year so that the plotted values show only the active range of variation.**





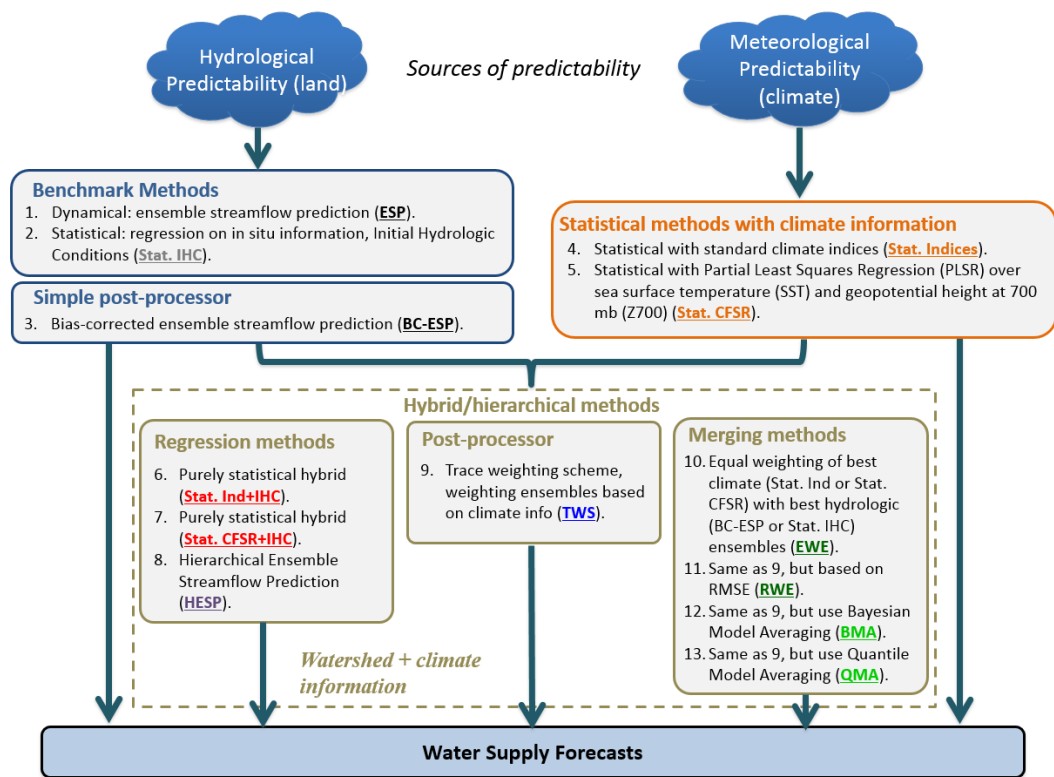

**Figure 3: Schematic figure showing all seasonal streamflow forecasting methods included in the inter-comparison framework. The benchmark methods are operationally implemented in the Western United States, and they are solely based on hydrologic predictability.**



**Figure 4: Monthly streamflow simulations (red) and observations (black) for the period Oct/1980 – Sep/2000. Left panels display monthly time series, with NSE and *r* denoting the Nash-Sutcliffe efficiency and correlation, respectively. Right panels show simulated and observed seasonal streamflow cycles. Results are displayed for (a) Dworshak Reservoir inflow (DWRI1); (b) Howard Hanson reservoir inflow (HHDW1); (c) Hungry Horse reservoir inflow (HHWM8); (d) Libby dam inflow (LYDM8); and (e) Prineville reservoir inflows (PRVO).**





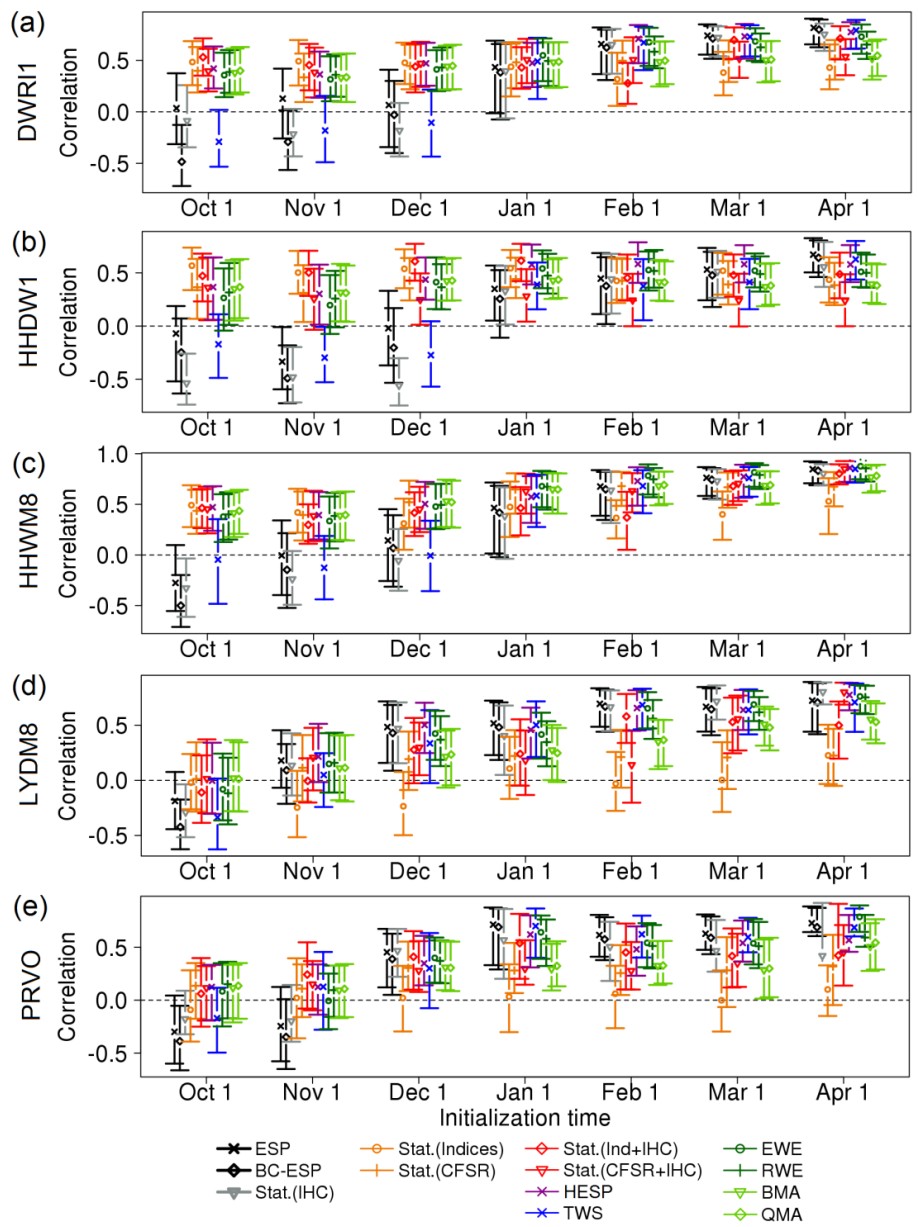

808

**Figure 5: Correlation coefficients of forecast ensemble medians versus observations obtained from all methods at different initialization dates. The error bars define 95% confidence limits obtained through bootstrapping with replacement. Results are displayed for (a) Dworshak Reservoir inflow (DWRI1); (b) Howard Hanson reservoir inflow (HHDW1); (c) Hungry Horse reservoir inflow (HHWM8); (d) Libby dam inflow (LYDM8); and (e) Prineville reservoir inflows (PRVO).**





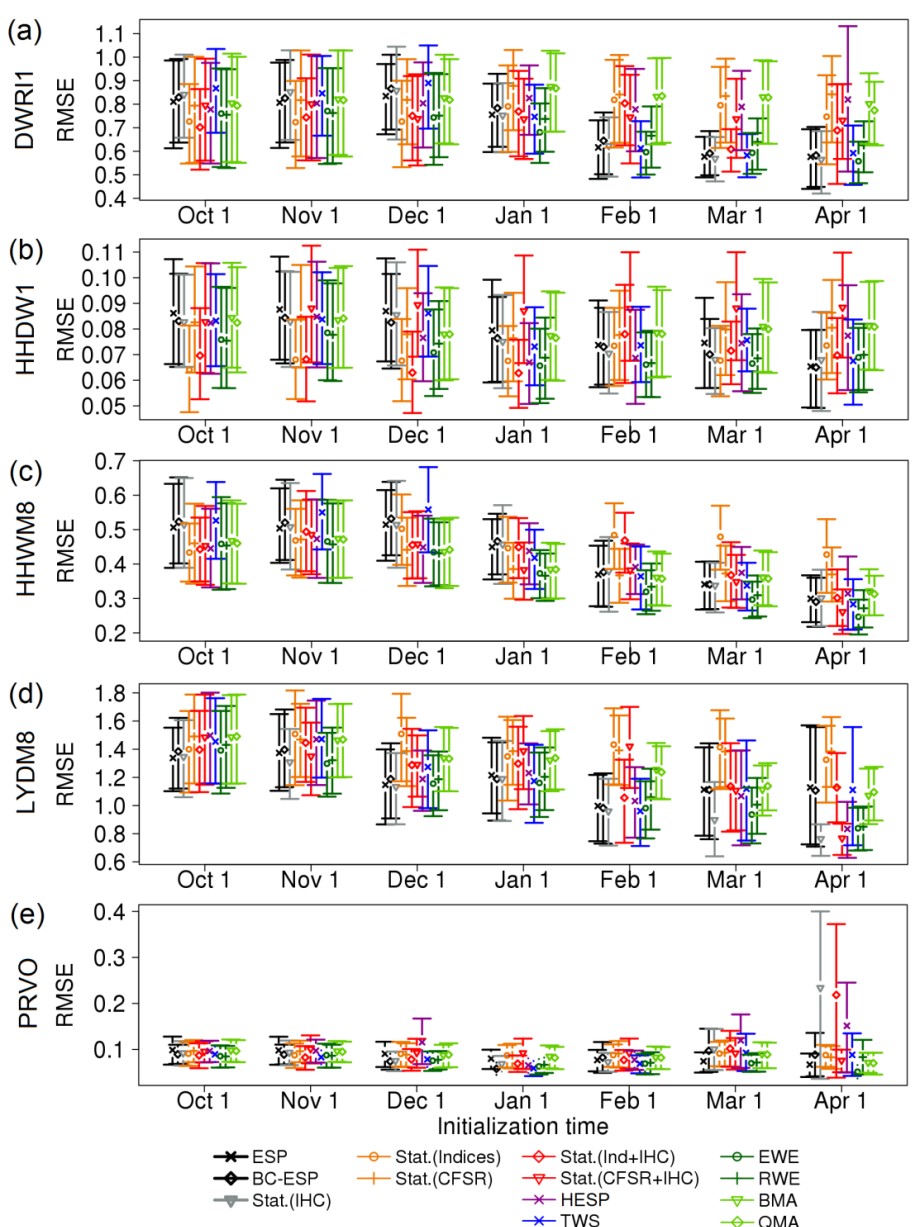


**Figure 6: Same as in Figure 5, but for root mean squared error (RMSE) of ensemble forecast medians versus observations.**
**See text for further details.**





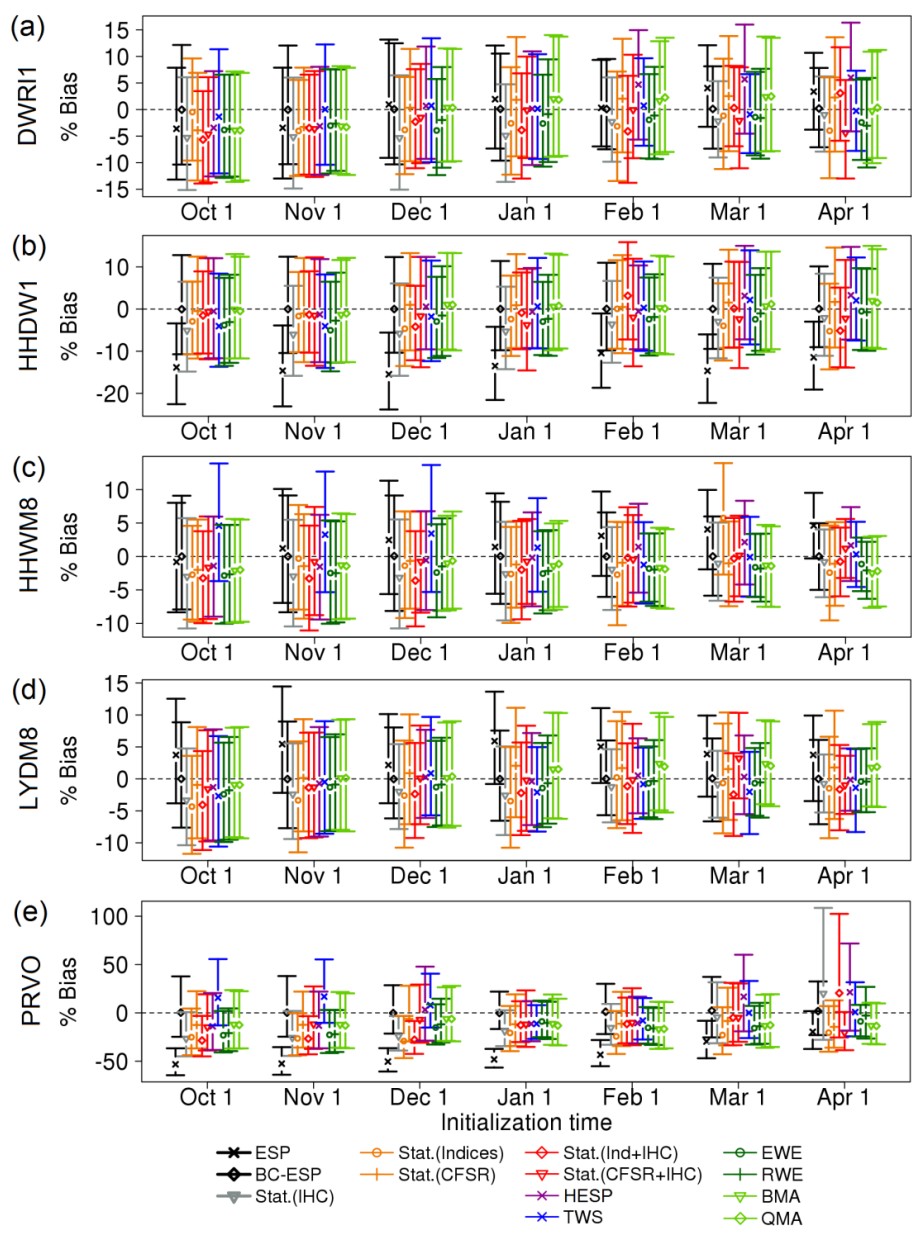


**Figure 7: Same as in Figure 5, but for percent bias (% bias) in forecast ensemble medians versus observations. See text for**
**further details.**





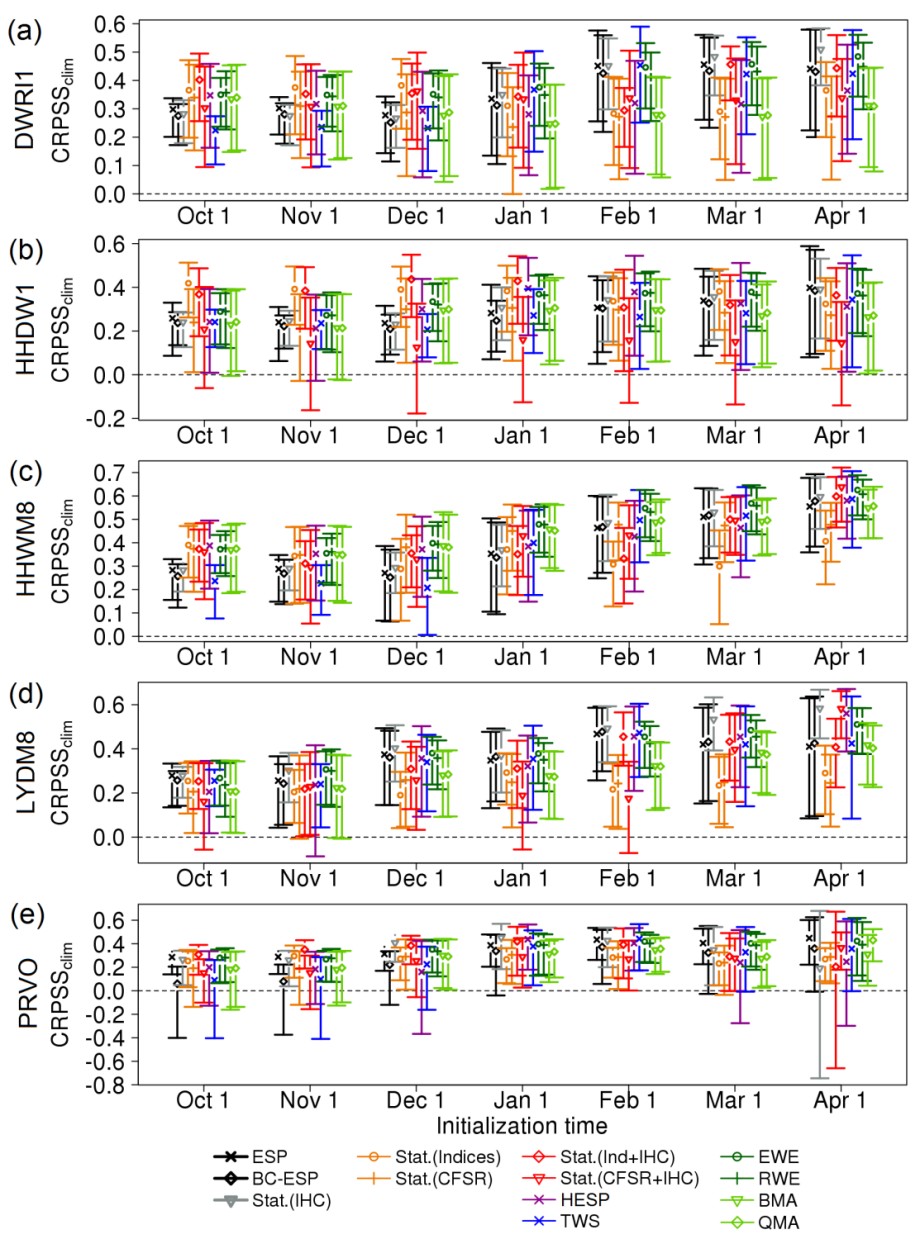


**Figure 8: Continuous Ranked Probability Skill Score of the forecast ensembles with respect to mean observed climatology (CRPSS$_{clim}$). See text for further details.**



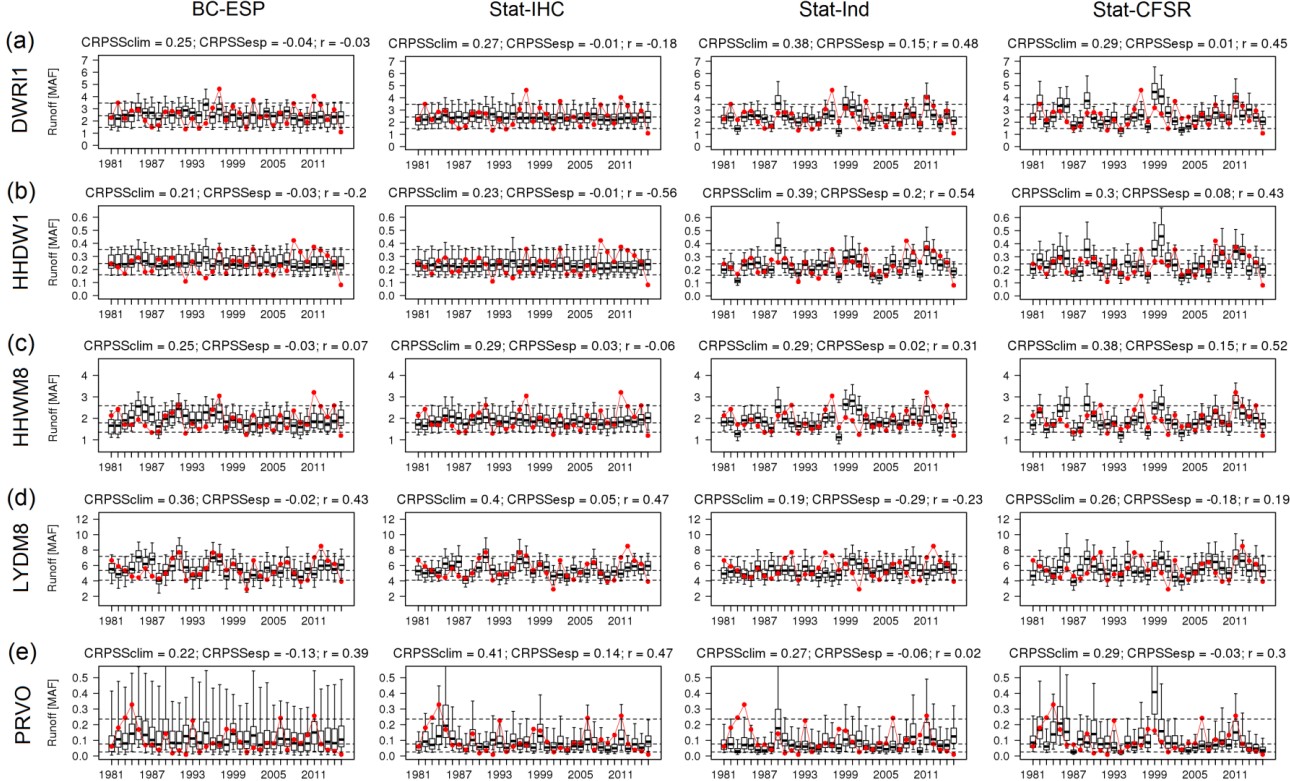


**Figure 9: Time series with cross-validated hindcasts initialized on December 1, obtained with two watershed-based methods (BC-ESP and Stat-IHC) and two climate-based techniques (Stat-Ind and Stat-CFSR) for the five case study locations (a-e). The verification metrics CRPSS$_{clim}$ and CRPSS$_{esp}$ denote continuous ranked probability skill scores using the mean climatology and raw ESP output as the reference, respectively. Black dashed lines represent 10%, 50% and 90% flows from the observed climatology, and boxplots show the 10$^{th}$, 30$^{th}$, 50$^{th}$, 70$^{th}$ and 90$^{th}$ hindcast percentiles.**



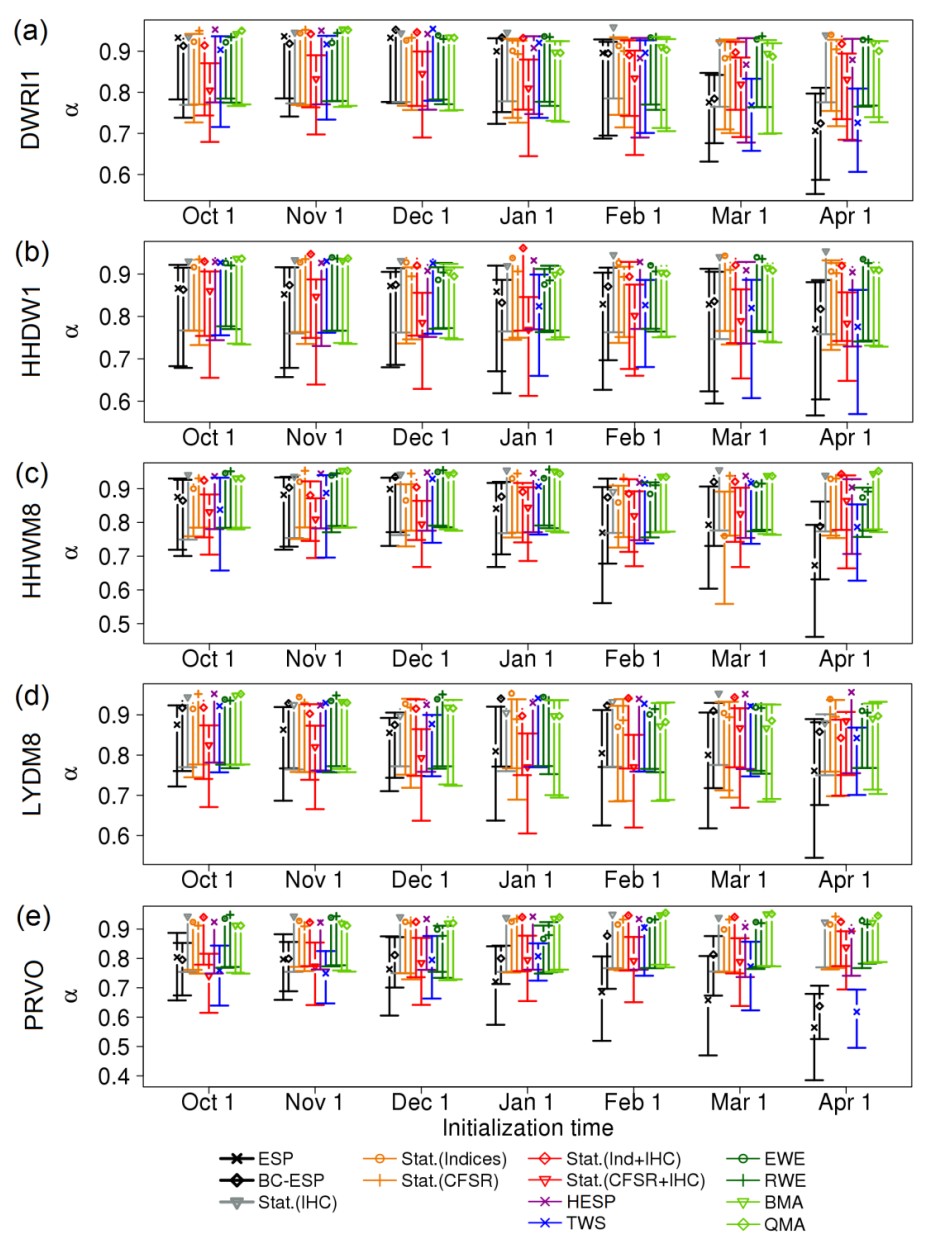

829

**Figure 10: The α reliability index for the hindcast ensembles for five case study locations. See text for further details.**



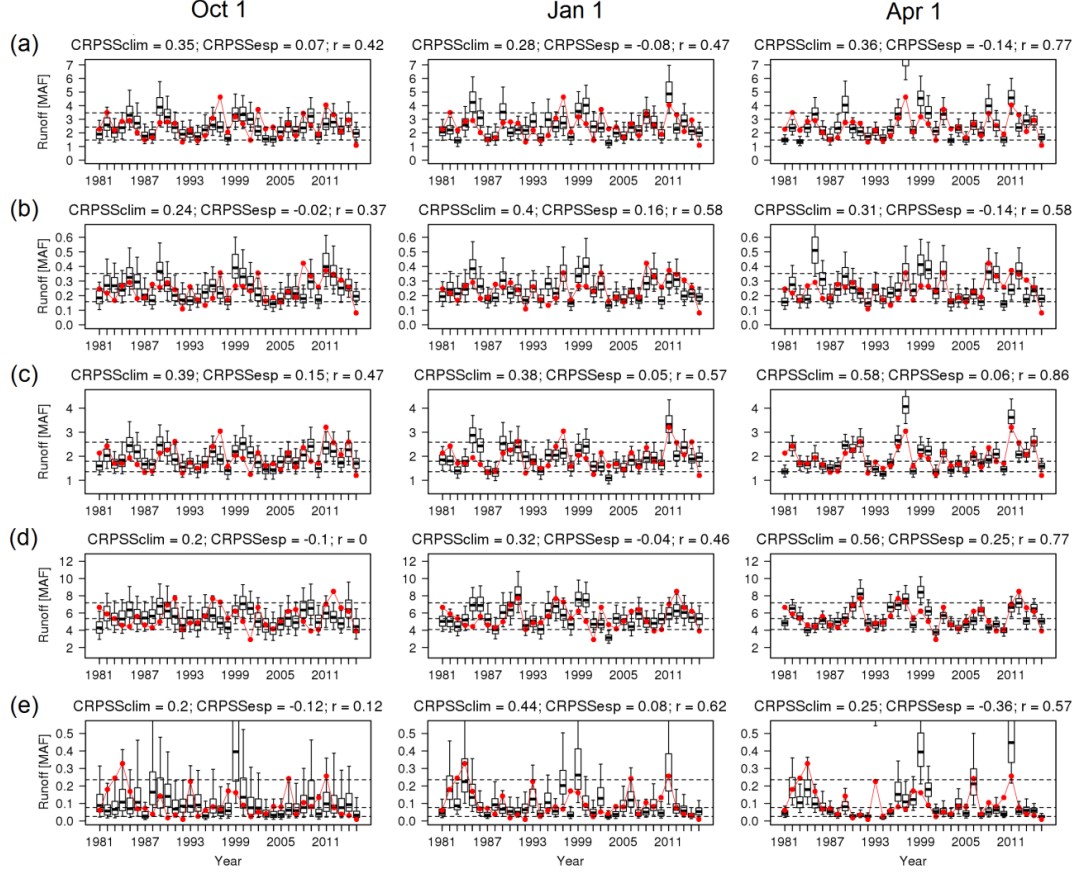

**Figure 11: Time series with cross-validated hindcasts obtained with the Hierarchical Ensemble Streamflow Prediction (HESP) approach, initialized on (left) October 1, (center) January 1, and (right) April 1. Results are displayed for the five case study locations: (a) Dworshak Reservoir inflow (DWRI1); (b) Howard Hanson reservoir inflow (HHDW1); (c) Hungry Horse reservoir inflow (HHWM8); (d) Libby dam inflow (LYDM8); and (e) Prineville reservoir inflows (PRVO). Black dashed lines represent 10%, 50% and 90% flows from the observed climatology, and boxplots show the 10th, 30th, 50th, 70th and 90th hindcast percentiles.**

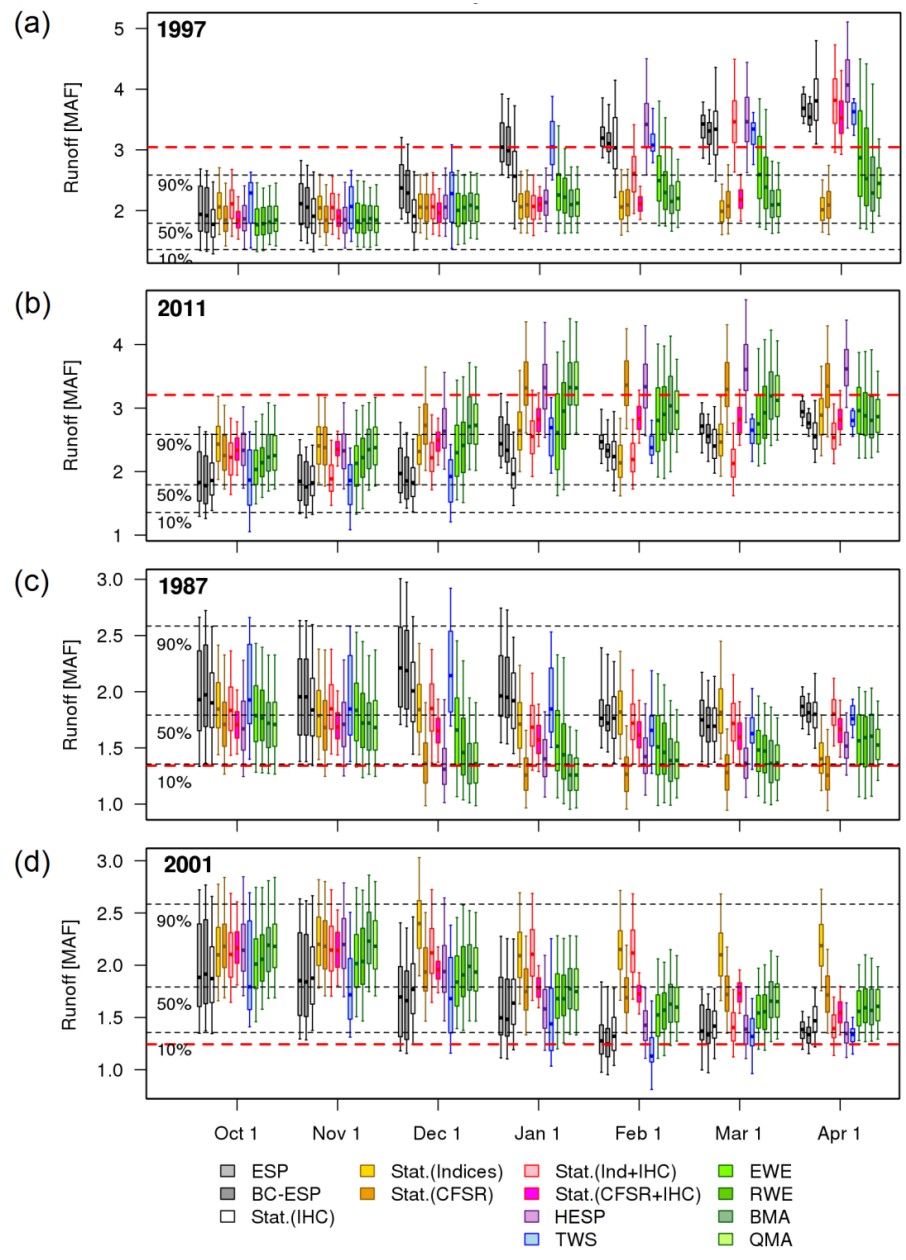

840

**Figure 12: April-July water supply forecasts obtained at the Hungry Horse reservoir (HHWM8) with different methods for two wet years – (a) 1997, and (b) 2011 – and two dry years – (c) 1987, and (d) 2001. The red dashed line represents the observed flow, while black dashed lines represent 10%, 50% and 90% flows from observed climatology, and boxplots show the 10th, 30th, 50th, 70th and 90th hindcast percentiles.**

845