# Peer review of "An intercomparison of approaches for improving predictability"

_Hydrology and Earth System Sciences, 2017_

## Referee Comment (RC1) · T. Pagano (Referee) · 21 Mar 2017

This is an interesting and well written comprehensive evaluation of over a dozen statistical, dynamical and hybrid seasonal streamflow forecasting techniques. The evaluation is done for about 20 years of 5 reservoirs in mostly snow-dominated climates of the Pacific Northwest US. My suggestions for changes are minor at best, with detailed comments below:

Title: I don't think "predictability" is the right word for the title. It implies something that's immutable and intrinsic, in the sense of theoretical maximum predictability, which is not something that could be "improved". Predictive skill of certain techniques or a forecasting enterprise can be improved, however.

Line 57 "current operational practice in the US still takes little to no advantage of largescale climate information for realtime seasonal streamflow forecasting" and later line 64-65 "these [operational] approaches rely solely on the predictability of [initial hydrologic conditions] and do not leverage any type of large-scale current or future climate information". From my experience as a forecaster, there were only very limited locations and leadtimes where the climate information provided substantial benefits. Things like El Nino indices were used in pacific northwest and southwest US for early (i.e. January) and pre-season (i.e. October-December) forecasts. I think it's strong to say that there was no use at all of climate information.

Line 89 Following the list of statistical water supply forecasting techniques. It may be useful to include in that list http://onlinelibrary.wiley.com/doi/10.1111/j.1752-1688.2009.00321.x/abstract because it also includes z-score regression and describes operational products.

Line 172 The universal use of the log transform on all the predictands. Operationally, forecasters use linear, square root, cube root and log transform statistical models, with log being the most extreme. The use of log everywhere wouldn't have been my first choice, and is probably responsible for "forecast blowouts" like 1993 in the Apr-1 / e panel on figure 11 (far lower right corner, only the lower whisker is visible on the chart). But since it's applied the same everywhere, it means that the intercomparison is valid in a relative rather than absolute sense. You might reassure the reader that you tried other transforms and the results were insensitive.

Line 261 The use of stepwise approach to model building. I think what you're describing here is the case where El Nino is predicting fall precipitation, and by the time January 1 comes around the precipitation is "in the bank" and so continued use of El Nino as a seasonal streamflow predictor after January 1 is redundant, if the equation also includes IHC variables. This was a common operational challenge in the US and a frustration to forecasters.

HESSD
Line 379 I think you find that El Nino provides a small amount of pre-HESSD dictability in October-December and by 1 January comes, initial hydrologic conditions are comparable to El Nino skill, but then by 1 Februlater. IHC are heavily dominant. This is consistent with ary and Interactive http://citeseerx.ist.psu.edu/viewdoc/download?doi=10.1.1.177.3158&rep=rep1&type=pdf https://scholar.google.com.au/citations?view op=view citation&hl=en&user=5hdY14AAAAAJ&citationcfonview=5hdY14A For many years 1 February was the start of the operational forecasting season and so there is little surprise that hydrologists were underwhelmed with what El Nino had to offer them. It wasn't until leadtimes were pushed back to 1 January, and then back to 1 October that hydrologists became more operationally interested.

Line 410 On explanations of why ESP is under-dispersive- The common way of explaining this is that NWS-style ESP does not consider parameter, data or model uncertainty, only uncertainty of future forcings.

Line 477 Generating custom climate indices beyond El Nino, creates useful information. I feel like this contradicts the statements on lines 383-385 where you say that this technique was the worst performer.

---

## Referee Comment (RC2) · B. Klein (Referee) · 21 Mar 2017

The manuscript compares different methods for seasonal water supply forecasts in several catchments in the Pacific Northwest region of the US. A large variety of different models was applied: purely statistical methods, methods based on watershed modelling as well as hybrid approaches using initial hydrologic conditions and / or climate information as input. Additionally different post-processing and merging methods are tested. The snow-dominated test catchments cover a wide range of hydrometeorological conditions and different atmospheric teleconnection signals.

The literature review of the most commonly used methods in seasonal streamflow forecasting is exhaustive and the results are nicely presented and compared. The real

value of this study is the comparison of a large variety of methods based on a common hindcast/verification framework using rigorous three years out cross validation. Using such a common framework an objective comparison of the performance of the different methods is possible. The paper is well written and should be foreseen for publication in HESS after minor revisions.

- You should probably use SI units instead of KCFS (Thousands of Cubic Feet Per Second) and MAF (Million acre Feet) (Standard in HESS)

P 5 line 155: missing first three year period in the brackets could be confusing why it is missing, add period: "... (e.g., 1981-1983, 1984-1986, 1987-1989, 1990-1992,etc.),..."

P 5 line 171: "...predictant data are normalized before ..." what do you mean by normalizing in this context? I assume z-scores are calculated or do you apply a normalization method such as box-cox? Please specify! Explain why you have used log-transformation of the predictant data and no other transformation method (e.g. Box-Cox, ...).

P 6 line 201: If the predictant was normalized (subtracting its expected value and dividing the difference by its standard deviation) before as stated above, the predictant has to be multiplied with the standard deviation and the mean has to be added before exponentiation. Is this correct? In this case the explanation of this procedure should be added.

P 7 line 210: Replace MRL with MLR

P 7 line 213: "... preciding seasonal predictor average and seasonal streamflow volume..." Is the MLR applied to log-transformed streamflow? Do you normalize the climate indices? Please specify!

P 8 line 246: Re-transformation of predicted streamflow should be added as additional step

P 9 line 284: Please explain shortly how the weighted resampling using the weights

1/RMSE works.

P 24 Table 1: In the table and in the main section the abbrevation RE (runoff efficiency) is used, in the table caption runoff ratio RR is used, please harmonize

P 34 and p 36: Please add explanation of red line (observation?) to figure caption

---

## Author Response (AR1)

**Replies to Referee #1**

**"An intercomparison of approaches for improving predictability in operational seasonal streamflow forecasting"**

Pablo A. Mendoza, Andrew W. Wood, Elizabeth Clark, Eric Rothwell, Martyn P. Clark, Bart Nijssen, Levi D. Brekke, and Jeffrey R. Arnold

We thank this reviewer for his time in commenting on our paper. We provide responses to each individual point below. For clarity, comments are given in italics, and our responses are given in plain text.

This is an interesting and well written comprehensive evaluation of over a dozen statistical, dynamical and hybrid seasonal streamflow forecasting techniques. The evaluation is done for about 20 years of 5 reservoirs in mostly snow-dominated climates of the Pacific Northwest US. My suggestions for changes are minor at best, with detailed comments below.

We are very pleased that this reviewer appreciates the contributions of this study.

*Title: I don't think "predictability" is the right word for the title. It implies something that's immutable and intrinsic, in the sense of theoretical maximum predictability, which is not something that could be "improved". Predictive skill of certain techniques or a forecasting enterprise can be improved, however.*

The reviewer makes a good point. To avoid confusion on the concept of "predictability", we have modified the title to "An intercomparison of approaches for improving operational seasonal streamflow forecasts" (L1-2).

Line 57 "current operational practice in the US still takes little to no advantage of largescale climate information for realtime seasonal streamflow forecasting" and later line 64-65 "these [operational] approaches rely solely on the predictability of [initial hydrologic conditions] and do not leverage any type of large-scale current or future climate information". From my experience as a forecaster, there were only very limited locations and leadtimes where the climate information provided substantial benefits. Things like El Nino indices were used in pacific northwest and southwest US for early (i.e. January) and pre-season (i.e. October-December) forecasts. I think it's strong to say that there was no use at all of climate information.

We have modified the text to reflect the reviewer's experience on this topic, though in truth the vast majority of statistical forecasting locations in the western US do not use climate indices to our knowledge – even in the Pacific Northwest (PNW). The paragraph now reads (L59-69):

"Despite generally promising findings from this body of work and from a number of agency development efforts (Weber et al., 2012; Demargne et al., 2014), the use of large-scale climate information for real-time seasonal streamflow forecasting in the US remains rare. In the western United States, where snowmelt commonly dominates the annual cycle of runoff, official WSFs are produced via two main approaches: (i) statistical models leveraging in situ watershed moisture measurements such as snow water equivalent (SWE), accumulated precipitation and streamflow (Garen, 1992; Pagano et al., 2004); and (ii) outputs from the National Weather Service (NWS) Ensemble Streamflow Prediction method (ESP; Day, 1985), which is based on watershed modeling. For the overwhelming majority of forecast locations, these approaches rely solely on the

predictability from IHCs (measured or modelled). A small number of locations can be found, however, where climate indices also serve as predictors in the statistical framework, and the NWS has recently implemented techniques through which climate model forecasts may eventually be applied to ESP (Demargne et al., 2014)"

Line 89 Following the list of statistical water supply forecasting techniques. It may be useful to include in that list http://onlinelibrary.wiley.com/doi/10.1111/j.1752-1688.2009.00321.x/abstract because it also includes z-score regression and describes operational products.

Certainly! This was an oversight as we are aware of that work, thus we have included the aforementioned reference, following the reviewer's suggestion (L94).

Line 172 The universal use of the log transform on all the predictands. Operationally, forecasters use linear, square root, cube root and log transform statistical models, with log being the most extreme. The use of log everywhere wouldn't have been my first choice, and is probably responsible for "forecast blowouts" like 1993 in the Apr-1 / e panel on figure 11 (far lower right corner, only the lower whisker is visible on the chart). But since it's applied the same everywhere, it means that the intercomparison is valid in a relative rather than absolute sense. You might reassure the reader that you tried other transforms and the results were insensitive.

We regret that we did not try other transformations as we were focused on relative outcomes, though this would have been a reasonable thing to do. In truth, we did place a great deal of importance on the transformation when the work was done, though since then our interactions with CSIRO has opened our eyes to the variation in the effectiveness of different transformations (including, for instance, the log-sinh). We do not have the bandwidth to go back and explore this issue, but for now we will highlight it for the readers based of the text of the comment. Hence, we have added the following sentences (L179-182):

"In practice, forecasters use a variety of transforms such as linear, square root, cube root, log and log-sinh (Wang et al. 2012). We did not explore alternative transforms, using the log consistently throughout, but recognize that the choice of transform can affect the quality of the forecast."

Line 261 The use of stepwise approach to model building. I think what you're describing here is the case where El Nino is predicting fall precipitation, and by the time January 1 comes around the precipitation is "in the bank" and so continued use of El Nino as a seasonal streamflow predictor after January 1 is redundant, if the equation also includes IHC variables. This was a common operational challenge in the US and a frustration to forecasters.

The reviewer correctly identifies the motivation for the technique, in the sense that HESP intends to handle seasonally varying sources of predictability separately, applying the climate predictors only to the portion of the flow variation that has not already been explained by the IHCs, if possible. If the signal from watershed moisture conditions becomes strong and is redundant,  $\varepsilon_{climate} = Q - f(IHC)$  (i.e., the residual from that relationship) cannot be explained robustly by climate information and HESP just defaults to Stat-IHC.

Line 379 I think you find that El Nino provides a small amount of predictability in October-December and by 1 January comes, initial hydrologic conditions are comparable to El Nino skill, but then by 1 February and later, IHC are heavily dominant. This is consistent with http://citeseerx.ist.psu.edu/viewdoc/download?doi=10.1.1.177.3158&rep=rep1&type=pdf https://scholar.google.com.au/citations?view\_op=view\_citation&hl=en&user=5hdY14AAAAAJ&ci tation\_for\_view=5hdY14AAAAAJ:YsMSGLbcyi4C For many years 1 February was the start of the operational forecasting season and so there is little surprise that hydrologists were underwhelmed with what El Nino had to offer them. It wasn't until leadtimes were pushed back to 1 January, and then back to 1 October that hydrologists became more operationally interested.

Indeed, our skill plots (Figures 5 and 8) align with the findings by Pagano and Garen (2006) and other researchers as to this point, specifically with the progression of seasonal streamflow forecast skill provided in their Figure 1. We thank the reviewer for this observation about the original initialization dates in February, which is encouraging if it indicates a trend toward even earlier start times where the climate information is relatively more important. We add the following sentence (L400-401):

"This progression of relative predictabilities from climate and watershed moisture conditions (Figures 5 and 8) is consistent with previous findings for the region (e.g., Pagano and Garen 2006)."

Line 410 On explanations of why ESP is under-dispersive- The common way of explaining this is that NWS-style ESP does not consider parameter, data or model uncertainty, only uncertainty of future forcings.

This is a good point. Indeed, ESPs are particularly under-dispersive at late forecast initializations, when uncertainty in IHCs dominates the total streamflow forecast uncertainty (Wood and Schaake 2008). We thank the reviewer for this observation, and have modified the text accordingly (L432-435):

"For such lead times, the uncertainty in ESP streamflow forecasts is underestimated due to reliance on a single modeled IHC that does not account for modeling errors (Wood and Schaake 2008), such that forecast spread derives only from uncertainty represented by the ensemble of future forcings."

Line 477 Generating custom climate indices beyond El Nino, creates useful information. I feel like this contradicts the statements on lines 383-385 where you say that this technique was the worst performer.

The reviewer refers to a comparison between the three hybrid regression techniques (Stat-Ind-IHC, Stat-CFSR-IHC, and HESP) in terms of probabilistic skill. We did find that for some basins (e.g., Dworshak and Hungry Horse) Stat-CFSR provides higher skill than using custom climate indices (Stat-Ind), outperforming also benchmark techniques at early initializations. Nevertheless, our results also show that when custom-indices are used in combination with stronger predictors, attempting to explain smaller amounts of variance, they are not as robust as using standard climate indices. We add a sentence to help explain this context (L407-409):

"When used in combination with other, stronger predictors, the parameter estimation cost of the CFSR-PLSR relative to an off-the-shelf index may be more exposed (leading to greater shrinkage of skill after cross-validation)."


We thank this reviewer for his time in commenting on our paper. We provide responses to each individual point below. For clarity, comments are given in italics, and our responses are given in plain text.

The manuscript compares different methods for seasonal water supply forecasts in several catchments in the Pacific Northwest region of the US. A large variety of different models was applied: purely statistical methods, methods based on watershed modelling as well as hybrid approaches using initial hydrologic conditions and / or climate information as input. Additionally, different post-processing and merging methods are tested. The snow-dominated test catchments cover a wide range of hydrometeorological conditions and different atmospheric teleconnection signals.

The literature review of the most commonly used methods in seasonal streamflow forecasting is exhaustive and the results are nicely presented and compared. The real value of this study is the comparison of a large variety of methods based on a common hindcast/verification framework using rigorous three years out cross validation. Using such a common framework an objective comparison of the performance of the different methods is possible. The paper is well written and should be foreseen for publication in HESS after minor revisions.

We are very pleased that this reviewer appreciates the contributions of this study.

You should probably use SI units instead of KCFS (Thousands of Cubic Feet Per Second) and MAF (Million acre Feet) (Standard in HESS)

We appreciate this sentiment. However, we consider that the paper would have much more value if the results are presented in units that are familiar to water managers in the US, since this is basically a water supply forecasting study. Therefore, we kindly ask the Editor and Publisher for permission to preserve the current flow units throughout the manuscript. We have added a short sentence to prepare readers for non-metric units (L199-L200):

"In this paper, results are reported in non-metric units because of their greater familiarity to readers from the US water management community."

Additionally, we have added SI units where possible, in parenthesis (L372-373, L377).

*P* 5 line 155: missing first three year period in the brackets could be confusing why it is missing, add period: "... (e.g., 1981-1983, 1984-1986, 1987-1989, 1990-1992, etc.),..."

We have added the first three-year period (1981-1983), following the reviewers' suggestion, and taken out the part of the series after the first two (L160).

*P* 5 line 171: "...predictant data are normalized before ..." what do you mean by normalizing in this context? I assume z-scores are calculated or do you apply a normalization method such as box-cox? Please specify!

The reviewer is correct: z-scores are computed using  $z = (x - \mu)/\sigma$ , where x represents the original variable, and  $\mu$  and  $\sigma$  represent the mean and standard deviation of the population, respectively. We clarify this procedure in the revised manuscript (L174-179).

*Explain why you have used log-transformation of the predictant data and no other transformation method (e.g. BoxCox, ...).*

We regret that we did not try other transformations as we were focused on relative outcomes, though this would have been a reasonable thing to do. In truth, we did place a great deal of importance on the transformation when the work was done, though since then our interactions with CSIRO has opened our eyes to the variation in the effectiveness of different transformations (including, for instance, the log-sinh). We do not have the bandwidth to go back and explore this issue, but for now we will highlight it for the readers based of the text of the comment. Hence, we have added the following sentences (L179-182):

"In practice, forecasters use a variety of transforms such as linear, square root, cube root, log and log-sinh (Wang et al. 2012). We did not explore alternative transforms, using the log consistently throughout, but recognize that the choice of transform can affect the quality of the forecast."

*P* 6 line 201: If the predictand was normalized (subtracting its expected value and dividing the difference by its standard deviation) before as stated above, the predictand has to be multiplied with the standard deviation and the mean has to be added before exponentiation. Is this correct? In this case the explanation of this procedure should be added.

The reviewer is correct, and the text has been modified to reflect this (L207, L211).

*P* 7 line 210: Replace MRL with MLR

We have corrected the text (L220), following the reviewer's suggestion.

*P* 7 line 213: "... predicting seasonal predictor average and seasonal streamflow volume..." Is the MLR applied to log-transformed streamflow? Do you normalize the climate indices? Please specify!

The reviewer is correct: MLR is applied to log-transformed streamflow, and then both predictand (flow in log space) and predictors (e.g., climate indices) are normalized. The general procedure used in this paper for statistical approaches is clarified, following this and a previous comment from this reviewer (L174-179):

"In the statistical approaches, seasonal flows are log-transformed, and predictor and predictand data are normalized before training statistical method parameters or weights (i.e., z-scores are computed using  $z = (x - \mu)/\sigma$ , where x represents the original variable, and  $\mu$  and  $\sigma$  represent the mean and standard deviation of x, respectively). The statistical models were applied in log-standard-normal space for forecast generation, then predictands are transformed from z-scores to log space (i.e., apply  $x = z\sigma + \mu$ , with  $x = \log(Q)$ ), and finally transformed back to streamflow space".

**P 8 line 246: Re-transformation of predicted streamflow should be added as additional step**

The step suggested by the reviewer is actually done, and therefore it has been added as part of the method description (L257-258):

"Ensemble forecasts are transformed from z-scores to log space, and finally exponentiated for conversion to flow space".

**P 9 line 284: Please explain shortly how the weighted resampling using the weights 1/RMSE works.**

RWE performs a weighted resampling from the two forecast sources (i.e., the best climate-only and best watershed-only forecasts) according to their skill during the training period. I.e., two weights 1/RMSE are obtained, where RMSE the root mean squared error of the ensemble median. These weights are normalized to make them sum 1, and finally obtain the fraction of the new 500-member ensemble coming from each forecast source. For example, if the resulting normalized weights are 0.4 and 0.6 for the best climate-only and best watershed-only forecasts, respectively, the RWE ensemble will contain 200 and 300 members from each prediction. This explanation has been added to the text in the revised manuscript (L296-L301).

*P 24 Table 1: In the table and in the main section the abbreviation RE (runoff efficiency) is used, in the table caption runoff ratio RR is used, please harmonize*

We have replaced runoff efficiency (RE) by runoff ratio (RR) in Table 1. Thanks for noting this.

P 34 and p 36: Please add explanation of red line (observation?) to figure caption

Indeed, the red line represents the observed flow volumes. We have incorporated this information to the caption of Figures 9 (L869) and 11 (L878).

3Bureau of Reclamation, Boise, USA

5

9 4Bureau of Reclamation, Denver, USA

10 5Climate Preparedness and Resilience Programs, U.S. Army Corps of Engineers, Seattle, USA

11 anow at: Advanced Mining Technology Center (AMTC), Universidad de Chile, Santiago, Chile

12 \*Correspondence to: Pablo A. Mendoza (pablo.mendoza@amtc.uchilecolorado.educl)

13 Abstract. For much of the last century, forecasting centers around the world have offered seasonal streamflow 14 predictions to support water management. Recent work suggests that the two major avenues to advance seasonal 15 predictability are improvements in the estimation of initial hydrologic conditions (IHCs) and the incorporation of 16 climate information. This study investigates the marginal benefits of a variety of methods using IHC and/or climate 17 information, focusing on seasonal water supply forecasts (WSFs) in five case study watersheds located in the U.S. 18 Pacific Northwest region. We specify two benchmark methods that mimic standard operational approaches -19 statistical regression against IHCs, and model-based ensemble streamflow prediction (ESP) - and then 20 systematically inter-compare WSFs across a range of lead times. Additional methods include: (i) statistical 21 techniques using climate information either from standard indices or from climate reanalysis variables; and (ii) 22 several hybrid/hierarchical approaches harnessing both land surface and climate predictability. In basins where 23 atmospheric teleconnection signals are strong, and when watershed predictability is low, climate information alone 24 provides considerable improvements. For those basins showing weak teleconnections, custom predictors from 25 reanalysis fields were more effective in forecast skill than standard climate indices. ESP predictions tended to have 26 high correlation skill but greater bias compared to other methods, and climate predictors failed to substantially 27 improve these deficiencies within a trace weighting framework.- Lower complexity techniques were competitive 28 with more complex methods, and the hierarchical expert regression approach introduced here (HESP) provided a 29 robust alternative for skillful and reliable water supply forecasts at all initialization times. Three key findings from 30 this effort are: (1) objective approaches supporting methodologically consistent hindcasts open the door to a broad 31 range of beneficial forecasting strategies; (2) the use of climate predictors can add to the seasonal forecast skill 32 available from IHCs; and (3) sample size limitations must be handled rigorously to avoid over-trained forecast 33 solutions. Overall, the results suggest that despite a rich, long heritage of operational use, there remain a number of 34 compelling opportunities to improve the skill and value of seasonal streamflow predictions.

**35 1 Introduction**

36 The operational hydrology community has long grappled with the challenge of producing skillful seasonal 37 streamflow forecasts to support water supply operations and planning. Proactive water management has become critical for many regions in the world that are susceptible to water stress associated with the intensification of the 38 39 water cycle. Paradoxically, although we have seen important technological advances - including increased 40 computing power, the broader availability to climate reanalysis, forecasts and reforecasts, and more complex 41 process-based hydrologic models (Pagano et al., 2016), the skill of operational seasonal runoff predictions in the US, 42 termed water supply forecasts (WSFs), has shown little or no improvement over time (e.g., Pagano et al., 2004; 43 Harrison and Bales, 2016). Hence, there is both a scientific and practical need to understand the potential of new 44 datasets, modeling resources and methods to accelerate progress towards more skillful and reliable operational 45 seasonal streamflow forecasts.

46 There is general consensus in the research community on the main opportunities to improve seasonal 47 streamflow prediction skill (e.g., Maurer et al., 2004; Wood and Lettenmaier, 2008; Yossef et al., 2013). These 48 include improving knowledge of: (i) the amount of water stored in the catchment - hereinafter referred to as initial 49 hydrologic conditions (IHCs), and (ii) weather and climate outcomes during the forecast period. Our ability to 50 leverage the first predictability source (i.e., hydrologic predictability) depends on the accuracy of watershed 51 observations and models, including model input forcings (e.g., precipitation and temperature), process 52 representations, and the effectiveness of hydrologic data assimilation (DA) methods. Our ability to leverage the 53 second source (climate predictability) depends both on how well we can characterize and predict the state of the 54 climate and on how effectively we can incorporate this information into streamflow forecasting methods. This idea 55 has been explored in different frameworks using standard indices - e.g., Niño3.4, the Pacific Decadal Oscillation 56 (PDO) - and/or custom (i.e., watershed-specific) climate indices derived from climate reanalyses (e.g., Grantz et al., 2005; Bradley et al., 2015), or using seasonal climate forecasts to run hydrologic model simulations (e.g., Wood et 57 58 al., 2005; Yuan et al., 2013).

59 Despite generally promising findings from this body of work and from a number of agency development 60 efforts (Weber et al., 2012; Demargne et al., 2014), the use of large-scale climate information for real-time seasonal streamflow forecasting in the US remains rarecurrent operational practice in the US still takes little to no advantage 61 62 of large-scale elimate information for real-time seasonal streamflow forecasting.-Clear examples can be found i In 63 the western United States, a large where snowmelt dominated region commonly dominates the annual cycle of 64 65 
[revised manuscript text omitted]
 = \frac{\sum_{i=1}^{N} (\mathbf{q}_{m,i} - \overline{\mathbf{q}_{m}})(\mathbf{o}_{i} - \overline{\mathbf{o}})}{\sqrt{\sum_{i=1}^{N} (\mathbf{q}_{m,i} - \overline{\mathbf{q}_{m}})^{2}} \sqrt{\sqrt{\sum_{i=1}^{N} (\mathbf{o}_{i} - \overline{\mathbf{o}})^{2}}}$ | Deterministic metric that varies [-1,1] with a perfect score of 1. It measures the linear association between forecasts and observations independent of the mean and variance of the marginal distributions.                                           |  |  |
| %Bias                                                                        | Percent bias                                                                                                   | $\%Bias = \frac{\sum_{i=1}^{N} (\mathbf{q}_{m,i} - \mathbf{o}_i)}{\sum_{i=1}^{N} \mathbf{o}_i} \times 100$                                                                                                                                                            | Deterministic metric that varies $(-\infty, \infty)$ , with perfect score of 0. It measures the difference between the mean of the forecasts and the mean of observations.                                                                             |  |  |
| RMSE                                                                         | Root mean squared error                                                                                        | $RMSE = \sqrt{\frac{1}{N}\sum_{i=1}^{N}(q_{m,i}-o_i)^2}$                                                                                                                                                                                                              | Deterministic metric that varies $[0,\infty)$ , with perfect score of 0.                                                                                                                                                                               |  |  |
| CRPSS                                                                        | Continuous ranked probability skill score                                                                      | $CRPSS = 1 - \frac{CRPS_{fest}}{CRPS_{ref}}$                                                                                                                                                                                                                          | Probabilistic metric that varies $(-\infty, 1]$ , with perfect score of 1. It measures
the skill of CRPS relative to a reference forecast (Hersbach, 2000). CRPS
quantifies the difference between the cumulative distribution (CDF)             |  |  |
|                                                                              |                                                                                                                | $CRPS = \frac{1}{N} \sum_{i=1}^{N} \int_{-\infty}^{\infty} \left[ F(\mathbf{q}) - F_o(\mathbf{q}) \right]^2 dq$                                                                                                                                                       | function of a forecast ( F ), and the corresponding CDF of the observations $(F_o)$ .                                                                                                                                                           |  |  |
|                                                                              |                                                                                                                | $F_o(\mathbf{q}) = \begin{cases} 0, & q < o \\ 1, & q \ge o \end{cases}$                                                                                                                                                                                              |                                                                                                                                                                                                                                                        |  |  |
| α                                                                            | $\alpha$ reliability index                                                                                     | $\alpha = 1 - 2 \left[ \frac{1}{N} \sum_{i=1}^{N} \left  \mathbf{P}_i(\mathbf{o}_i) - U(\mathbf{o}_i) \right  \right]$                                                                                                                                                | Probabilistic metric that varies [0,1]. It quantifies the closeness between
the empirical CDF of sample p-values with the CDF of a uniform
distribution. A value of 0 is the worst, and 1 reflects perfect reliability
(Renard et al., 2010). |  |  |
| $q_{m,i}$ : Forecast ensemble median for year i .                     |                                                                                                                |                                                                                                                                                                                                                                                                       |                                                                                                                                                                                                                                                        |  |  |
| $\overline{\mathbf{q}_m}$ : Temporal average over forecast ensemble medians. |                                                                                                                |                                                                                                                                                                                                                                                                       |                                                                                                                                                                                                                                                        |  |  |
| $\mathbf{o}_i$ : Ob                                                          | $o_i$ : Observation for year i .                                                                        |                                                                                                                                                                                                                                                                       |                                                                                                                                                                                                                                                        |  |  |
| ō: Tei                                                                       | $\overline{\mathbf{o}}$ : Temporal average of observations.                                                    |                                                                                                                                                                                                                                                                       |                                                                                                                                                                                                                                                        |  |  |
| $P_i(o_i)$                                                                   | $\mathbf{P}_i(\mathbf{o}_i)$ : Non-exceedance probability of $o_i$ using ensemble forecasts at year i . |                                                                                                                                                                                                                                                                       |                                                                                                                                                                                                                                                        |  |  |

 $U_i(\mathbf{o}_i)$ : Non-exceedance probability of  $o_i$  using the uniform distribution U[0,1].